# α-amino trimethylation of CENP-A by NRMT is required for full recruitment of the centromere

Kizhakke M. Sathyan[1], Daniele Fachinetti[2] & Daniel R. Foltz[1,3,4]

Centromeres are unique chromosomal domains that control chromosome segregation, and are epigenetically specified by the presence of the CENP-A containing nucleosomes. CENP-A governs centromere function by recruiting the constitutive centromere associated network (CCAN) complex. The features of the CENP-A nucleosome necessary to distinguish centromeric chromatin from general chromatin are not completely understood. Here we show that CENP-A undergoes α-amino trimethylation by the enzyme NRMT *in vivo*. We show that α-amino trimethylation of the CENP-A tail contributes to cell survival. Loss of α-amino trimethylation causes a reduction in the CENP-T and CENP-I CCAN components at the centromere and leads to lagging chromosomes and spindle pole defects. The function of p53 alters the response of cells to defects associated with decreased CENP-A methylation. Altogether we show an important functional role for α-amino trimethylation of the CENP-A nucleosome in maintaining centromere function and faithful chromosomes segregation.

[1] Department of Biochemistry and Molecular Genetics, University of Virginia, Charlottesville, Virginia 22908, USA. [2] Department of Cell Biology and Cancer, Institut Curie, PSL Research University, CNRS, UMR 144, 26 rue d'Ulm, Paris 75005, France. [3] Northwestern University, Feinberg School of Medicine, Department of Biochemistry and Molecular Genetics, Chicago, Illinois 60611, USA. [4] Robert H. Lurie Comprehensive Cancer Center, Northwestern University Feinberg School of Medicine, Chicago, Illinois 60611, USA. Correspondence and requests for materials should be addressed to D.R.F. (email: dfoltz@northwestern.edu).

The centromere is evident as the primary constriction on each chromosome, and is necessary for the faithful segregation of chromosomes. In higher eukaryotes, the underlying DNA does not specify centromere position, and although most centromeric DNA is repetitive, the DNA sequence found at centromeres across different species is highly divergent[1]. Instead, centromeres are specified by a unique nucleosome in which the histone variant CENP-A replaces canonical H3. The CENP-A nucleosome is sufficient to direct the assembly of the constitutive centromere associated protein network (CCAN), and determine the site of kinetochore formation at an ectopic location[2–6]. During mitosis the CCAN components CENP-T and CENP-C recruit NDC80 and Mis12 complex to mediate the assembly of the mitotic kinetochore[7–9].

Members of the CCAN make multiple contacts with different domains of the CENP-A nucleosome to direct the CCAN to the centromere. CENP-C interacts directly with the carboxyl terminus of CENP-A within the context of the nucleosome[5,10–12] and is one way in which the CENP-A nucleosome is uniquely recognized by the CCAN complex. CENP-N interacts with the CATD domain within the histone fold of CENP-A [5,11,13]. The amino terminus of CENP-A contributes to the stabilization of CENP-B binding at human centromeres by direct interaction[14,15], and mediates the recruitment of CENP-T and CENP-C at ectopic sites in human cells and at endogenous centromere in *Schizosaccharomyces pombe*[4,16].

Posttranslational modifications of the unstructured N-terminal tail of H3 are major modifiers of nucleosome function[17]. Phosphorylation of the human CENP-A amino terminus occurs at a poorly conserved Aurora B consensus site (Ser. 7) and additionally at Ser16 and Ser18, which are well conserved across vertebrates[18–21]. Perturbation of CENP-A Ser16/18 phosphorylation leads to changes in the compaction of CENP-A containing chromatin *in vitro* and increased rate of chromosome missegregation[21]. Ser68 phosphorylation within the histone fold of CENP-A has also been reported to influence CENP-A nucleosome deposition[22].

The initiating methionine of CENP-A is removed shortly after synthesis and the resulting alpha-amino group of Gly1 is trimethylated[21]. Maximal methylation of CENP-A is observed on nucleosomal CENP-A during mitosis; although, methylation of CENP-A can be detected at other times in the cell cycle and on CENP-A before its incorporation into the nucleosome. Methylation of the α-amino group of proteins was described three decades ago and has been observed on a diverse group of proteins in humans[23]. The functional importance of amino-terminal methylation has been shown for a few of these proteins, including RCC1, CENP-B and DDB2 (refs 23–26). N-terminal RCC Methyl transferase 1 (NRMT1) was originally identified as the enzyme responsible for methylating RCC1 *in vivo*, but has shown to methylate CENP-B, DDB2 and CENP-A, *in vitro*. NRMT2 is a developmentally controlled gene and usually monomethylates proteins[27]. NRMT1 recognizes an amino acid motif of X$_{aa}$–Pro–Lys/Arg (X$_{aa}$ denotes small side chain amino acids) at the N-terminus as a methylation signature[28]. The loss of NRMT1 is associated with increased sensitivity to DNA damage and promotes tumorigenesis[29].

Here, we demonstrated that the enzyme NRMT1 methylates CENP-A both *in vivo* and *in vitro,* and that α-amino terminal trimethylation is an essential feature of the CENP-A tail. Expression of CENP-A mutants that lack methylation lead to lagging chromosomes and multipolar spindle formation in p53-deficient cancer cells due to centriole disengagement and/or centriolar splitting. Methylation mutants have reduced CENP-T and CENP-I localization at the centromere and impaired kinetochore function. Moreover, cells expressing CENP-A methylation mutants form larger colonies when tested by colony formation assay and form tumours faster in mouse xenografts, suggesting the phenotypes associated with unmethylated CENP-A provide a survival advantage for p53 deficient cancer cells. In summary, we have found a major role of α-amino trimethylation to maintain centromere function and faithful segregation of chromosomes.

## Results

**NRMT1 methylates CENP-A *in vivo*.** To determine whether NRMT1 plays a role in CENP-A methylation *in vivo* we developed a specific antibody against the methylated CENP-A amino terminus. We assessed the specificity of this antibody using an *in vitro* methylation assay[21,23]. *In vivo*, the initiating methionine of CENP-A is removed by methionine aminopeptidase leaving the Glycine (Gly1) α-amino group available for methylation (Fig. 1a)[21]. To recapitulate the methionine-cleaved amino terminus of CENP-A we purified the initial 10 amino acids of CENP-A as an eGFP-fusion protein with 6XHis fused to CENP-A separated by Factor-X cleavage site on the N-terminus (Fig. 1b). The amino terminal Gly1 was exposed following Factor X cleavage and used as a substrate for *in vitro* methylation with recombinant NRMT1 (ref. 23). Western blot analysis shows an antibody raised against the methylated CENP-A peptide recognizes the *in vitro* methylated CENP-A but does not recognize the unmethylated CENP-A (Fig. 1c, Supplementary Fig. 1c–d). Pre-incubating the antibody with the methylated CENP-A peptide, or knockdown of CENP-A by shRNA, completely abolished centromere staining with the methylation specific antibody (Supplementary Fig. 1a–b). To determine whether NRMT1 is the enzyme responsible for methylation of CENP-A *in vivo*, we suppressed NRMT1 expression by shRNA treatment in HeLa cells stably expressing an eGFP-tagged CENP-A. CENP-A methylation was completely lost when NRMT1 was suppressed, as shown by both western blotting and immunofluorescence (Fig. 1d,e). The same result was further observed when NRMT1 was suppressed in HeLa cells and stained for endogenous CENP-A (Fig. 1f) Therefore, NRMT1 methylates CENP-A both *in vivo* and *in vitro*.

**CENP-A α-amino methylation is an independent event.** To determine if there was crosstalk between posttranslational modifications of the CENP-A tail, we identified mutations of CENP-A that eliminated NRMT1 methylation of the amino terminus. Three CENP-A mutants were created that altered the NRMT1 consensus sequence (Fig. 1g). These mutants were tested using previously reported NRMT1 mediated *in vitro* methylation assay that employs tritiated SAM ($^3$H-S-adenosyl-methionine), as a radioactive methyl donor (Fig. 1h)[23]. All three CENP-A mutants that we expressed and purified were not methylated by NRMT1 in this assay.

Three eGFP-tagged CENP-A methylation mutants (MT1, MT2 and MT3) were stably expressed in HeLa cells. None of these mutants were detected by the methyl specific CENP-A antibody (Fig. 1i; Supplementary Fig. 2c). CENP-A mutants containing alanine substitutions of the three known phosphorylations of the tail at Ser6 (a.k.a. Ser7), Ser16 and Ser18 (refs 19–21) were all recognized by the anti-methylated CENP-A antibody to the same degree as wild-type CENP-A (Fig. 1i). Therefore, CENP-A α-amino methyation is not influenced by phosphorylation status of the CENP-A tail.

CENP-A alanine mutants of Ser6 or Ser16/Ser18 phosphorylation sites were expressed as eGFP fusions to the C-terminus of CENP-A. CENP-A phosophorylation is not affected by CENP-A methylation. CENP-A MT1 that was not methylated was readily

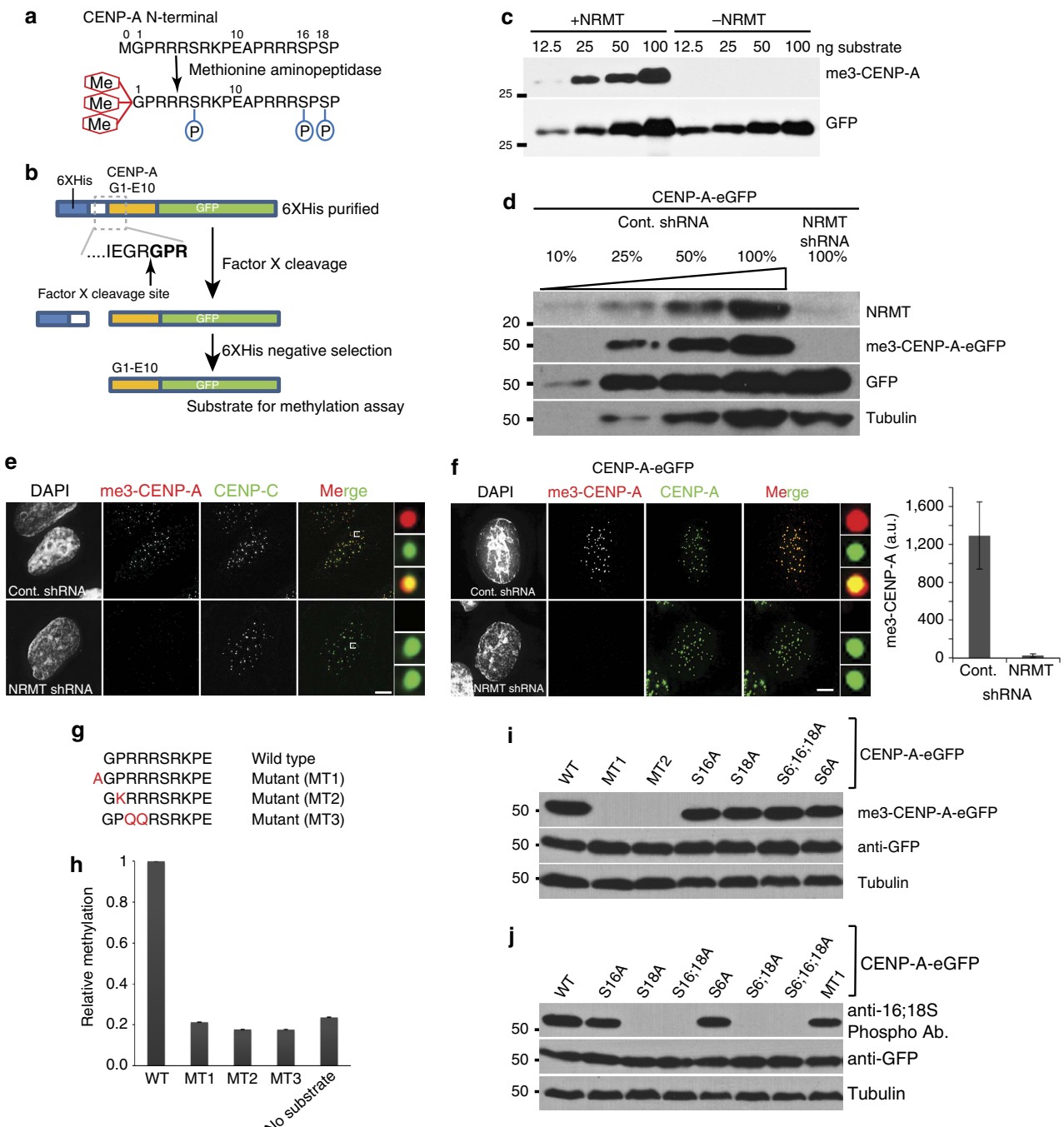

**Figure 1 | NRMT1 methylates CENP-A *in vitro* and *in vivo*. (a)** The initiating Methionine of CENP-A is removed by methionine aminopeptidase. Posttranslational modifications of the amino terminus are indicated: Me, α-amino-terminal methylation; P, phosphorylation. **(b)** Schematic of the first 10 amino acids of the CENP-A tail fused with GFP on the C-terminus and 6XHis on the N-terminus. Factor X cleavage exposes the N-terminal glycine for modification by NRMT. **(c)** Anti-me3-CENP-A antibody specifically recognizes CENP-A methylated *in vitro* by NRMT **(d)** Western blot of extracts from HeLa cells stably expressing CENP-A-eGFP in which NRMT was suppressed by shRNA shows a loss of CENP-A α-amino trimethylation. **(e)** Immunofluorescence analysis of the HeLa cell treated with NRMT1 shRNA using CENP-A me3 antibody shows loss of CENP-A methylation at centromeres. **(f)** Immunofluorescence using CENP-A me3 antibody of HeLa cell stably expressing CENP-A-LAP and treated with NRMT1 shRNA. Scale bar, 10 μm. Error bars indicate s.d. Experiment done in duplicates. **(g)** Amino acid sequence of the CENP-A mutants used in this study. **(h)** *In vitro* NRMT1 methylation assay using factor X cleaved CENP-A tail as a substrate in the presence of $^3$H-S-adenosyl-methionine. A filter-binding assay was used to determine the incorporation of radioactive methyl groups into CENP-A amino termini. The experiment was done in triplicate, $n = 3$. Error bars indicate s.d. **(i,j)** Immunoblot using the methyl specific CENP-A antibody of extracts from cells expressing CENP-A S16 and S18 phosphorylation mutants show that phosphorylation of S16 and S18 are not required for methylation of CENP-A. Likewise, mutations that eliminate CENP-A methylation do not affect the phosphorylation of CENP-A as shown by immunoblot with a phosphospecific CENP-A antibody.

recognized by an antibody that recognizes CENP-A phosphorylated on Ser 16 and Ser 18 (Fig. 1j; Supplementary Fig. 2a, b). Likewise, CENP-A mutants MT1 and MT2 did not affect S6 phosphorylation; although a third mutant (MT3) did alter S6 phosphorylation. This can be attributed to disruption of the Aurora B consensus site within the CENP-A tail (Supplementary Fig. 2c,d). All these data suggest that CENP-A methylation is independent of other posttranslational modifications within its N-terminal tail.

**CENP-A α-amino methylation is required for cell survival.** The carboxyl and amino terminal tails of CENP-A contain partially redundant functions, such that elimination of either does not significantly affect cell viability[14]. However, the loss of both tails leads to cell death and dramatic changes in the CCAN. To determine the contribution of CENP-A α-amino trimethylation to the essential function of the CENP-A amino terminus, we stably expressed CENP-A and the MT1 CENP-A methylation mutant under conditions where endogenous CENP-A could be completely removed. This was achieved using RPE-1 cells in which the only functional allele of *CENP-A* was flanked by LoxP sites (*CENP-A$^{-/F}$* cells)[14]. The final copy of the endogenous CENP-A gene can be removed from these cells by infection with adenovirus expressing Cre recombinase (Ad-Cre) (Fig. 2a,b).

RPE1 *CENP-A$^{-/F}$* cells stably expressing GFP-tagged CENP-A mutants were infected with Ad-Cre and their viability was assessed by colony formation assay after 14 days. There was no difference in viability between RPE-1 CENP-A$^{-/-}$ cells rescued with wild-type CENP-A ($^{WT}$CENP-A), and those expressing the $^{MT1}$CENP-A methylation mutant in the colony formation assay (Fig. 2a–d). This was expected based on previous results from Fachinetti *et al.* that showed the amino and carboxyl termini are partially redundant for viability[14]. To test the essential function of the amino terminus we created a CENP-A methylation mutant in which the C-terminal amino acids LEEGLG were replaced by the Histone H3 amino acids GERA ($^{MT1}$CENP-A$^{CH3}$). The $^{MT1}$CENP-A$^{CH3}$ chimera failed to form colonies, similar to cells not expressing a rescue construct (Fig. 2c,d). In contrast, methylated $^{WT}$CENP-A$^{CH3}$ was able to rescue cell viability. This is similar to what has been reported previously for the expression of the H3$^{CATD}$ chimeric protein containing the CENP-A targeting domain in the context of histone H3.1. These data suggest that CENP-A methylation is an essential feature of the CENP-A amino terminus required for cell growth.

**Methylation is required for localization of CENP-T and CENP-I.** To determine whether CENP-A methylation contributes to CCAN formation we examined the recruitment of CCAN proteins to the centromere in RPE1 *CENP$^{-/F}$* cells that were expressing eGFP tagged CENP-A wild type or methylation mutants containing the CENP-A or H3 carboxyl termini in

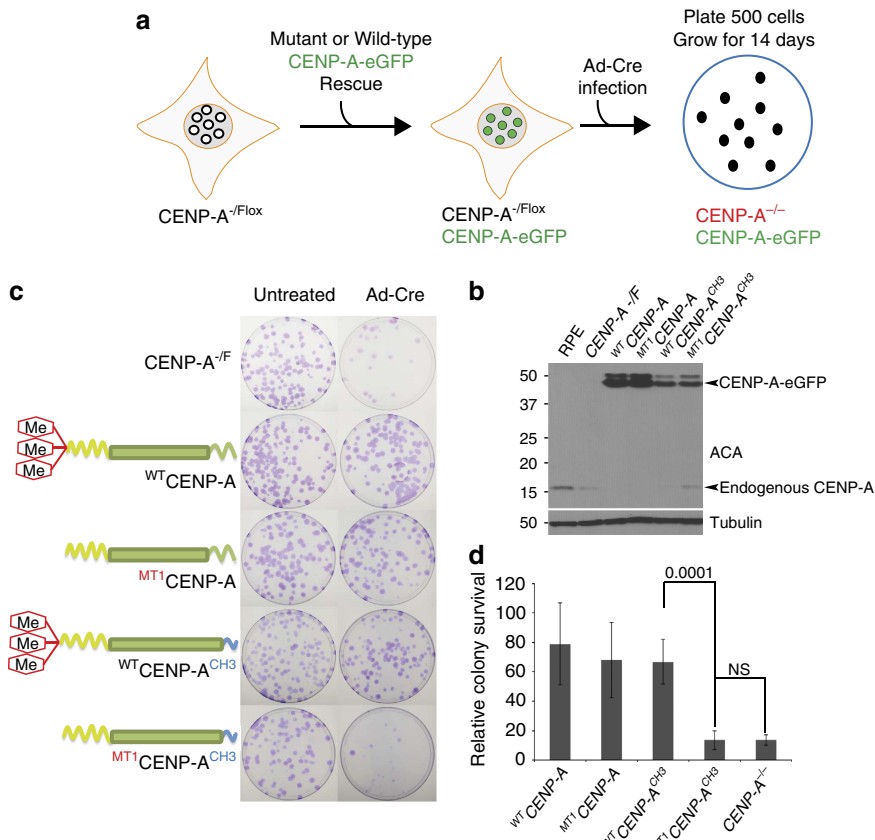

**Figure 2 | CENP-A α-amino methylation required for cell survival. (a)** Schematic diagram of the experiment. eGFP-tagged wild type and mutant CENP-A was stably integrated in RPE-CENP-A -/F cells by lentiviral transduction. The floxed allele of endogenous CENP-A was removed by infection with adenovirus encoding cre-recombinase. Five hundred cells were plated on 10 cm tissue plates and colony formation was monitored after 14 days. **(b)** Western blot showing expression of CENP-A wild type and mutant constructs in RPE-CENP-A −/F cells prior to Ad-Cre infection. **(c)** Representative examples of the colony formation assay. The cells carrying the CENP-A methylation mutant combined with C-terminal H3 swap showed lowest survival, comparable to CENP-A -/F cells not expressing a rescue construct. **(d)** Quantitation of relative colony survival. The experiments were done in triplicate, n = 3. Error bars indicate s.d. Indicated *P* values were determined by $\chi^2$ test.

interphase cells. There was no difference in the recruitment of the CCAN proteins CENP-B, CENP-C, CENP-I and CENP-T to centromeres between cells expressing wild-type CENP-A and CENP-A methylation mutants when endogenous CENP-A was present (Supplementary Fig. 3a–f).

However, when the endogenous CENP-A was removed by Cre-recombinase, we observed significant differences in CCAN recruitment after 5 days depending on the C-terminal sequence and alpha-amino terminal methylation status of CENP-A (Fig. 3a). CENP-C levels were significantly reduced in cells rescued with CENP-A containing the H3 C-terminus ($^{WT}$CENP-A$^{CH3}$) (Fig. 3b,d) consistent with a direct role for the C-terminus in recruiting CENP-C, as shown previously[5,14]. The amino terminus of CENP-A also plays a role in CENP-C recruitment via its binding with CENP-B (ref. 15). However, the methylation of CENP-A is not involved in this process, since we observed no difference in CENP-C recruitment between wild-type ($^{WT}$CENP-A) and CENP-A methylation mutants ($^{MT1}$CENP-A), irrespective of the C-terminal tail.

In contrast, CENP-T and CENP-I localization at the centromere was significantly reduced in CENP-A methylation mutants ($^{MT1}$CENP-A) relative to wild-type expressing cells ($^{WT}$CENP-A) (Fig. 3b–d). The reduction of CENP-T and CENP-I was observed in methylation mutants compared to the wild-type N-terminus when the CENP-A C-terminus was present ($^{WT}$CENP-A versus $^{MT1}$CENP-A), and when the H3 C-terminal chimeras were expressed ($^{WT}$CENP-A$^{CH3}$ versus $^{MT1}$CENP-A$^{CH3}$). Consistent with this result we found that cells rescued with a CENP-A mutant completely lacking its N-terminal tail ($^{ΔNH2}$CENP-A)[15] show a significant reduction in CENP-I and CENP-T localization at the centromere (Supplementary Fig. 3g). A reduction of CENP-I and CENP-T centromeric levels was also observed when comparing $^{WT}$CENP-A expressing cells with $^{WT}$CENP-A$^{CH3}$. Therefore CENP-T and CENP-I are recruited via two independent pathways, through the C-terminal tail—possibly through CENP-C—and by an N-terminus dependent pathway that requires alpha-amino terminal methylation. Similar decreases in CENP-T and CENP-I levels in the methylation mutant were observed in HeLa cells where endogenous CENP-A is suppressed by shRNA (Supplementary Fig. 3h–j).

To determine if the impairment of the CCAN caused by lack of CENP-A methylation reduces the kinetochore function, cells were treated with 0.1 μM nocodazole for 6 h and then released for 1 h to progress into anaphase and examined for lagging chromosomes (Fig. 3h). Cells expressing the CENP-A methylation mutant were significantly more likely to contain lagging chromosomes compared to wild type CENP-A replaced cells and parental RPE-1 cells (Fig. 3i,j). Furthermore we found that loss of CENP-A methylation resulted in a significantly higher number of micronuclei even without nocodazole treatment (Fig. 3k). Levels of the kinetochore protein NDC80 were also significantly reduced in the $^{MT1}$CENP-A mutant replaced cells compared to parental or CENP-A wild-type replaced cells (Fig. 3l,m). In summary, CENP-A methylation is required for CCAN formation and faithful chromosome segregation.

**Methylation is required for accurate chromosome segregation.** To further understand the role of CENP-A methylation in chromosome segregation, we developed HeLa cell lines stably expressing C-terminal eGFP-tagged CENP-A wild-type, or one of the methylation mutants (MT1-3). Doxycycline inducible CENP-A shRNA was integrated into these cells using lentiviral transduction to suppress endogenous CENP-A. Cells were treated with Dox for 8–15 days resulting in significant reduction in endogenous CENP-A protein level (Supplementary Fig. 4b).

When CENP-A was suppressed under these condition cells formed multipolar spindles in the mutant cells (Fig. 4a–c), similar to NRMT1 knockdown (Supplementary Fig. 4a,b)[23]. To further, support these observations we used three separate shRNA sequences and observed almost identical rates of multipolar spindle formation on CENP-A depletion suggesting that the phenotype was not due to off target effects of the shRNA (Fig. 4d–f). The multipolarity caused by shRNA suppression could be rescued by expression of wild-type CENP-A-eGFP (Fig. 4b,c). However, shRNA induced cells rescued with the CENP-A methylation mutants showed a comparable degree of multipolar spindle formation as non-rescued cells (Fig. 4b,c). This finding suggests that loss of methylation leads to a reduction in CENP-A function.

Failure in cytokinesis or centrosome reduplication cause increased centriole numbers[30]. We found that the majority of multipolar cells in both the CENP-A shRNA treated cells and those rescued with CENP-A methylation mutant had only four centrioles (Fig. 4g,h). The number of acentric spindle poles did not change between controls and CENP-A methylation mutants (Supplementary Fig. 4c,d). These findings indicate that multipolar spindles induced by CENP-A depletion or rescue using CENP-A methylation mutant were formed by either centriole disengagement or pericentriolar matrix dissociation and not by a failure in cytokinesis or by centrosome reduplication.

**p53 activity limits the formation of multipolar spindles.** To determine whether the p53 status of these cell lines may influence the formation of multipolar spindles when CENP-A is unmethylated we expressed wild-type and CENP-A methylation mutant in the $p53^{+/+}$ and $p53^{-/-}$ HCT116 cell lines[31]. These cells were further infected with Dox inducible CENP-A shRNA (sh1) (Fig. 5a,b). CENP-A suppression in HCT116 $p53^{-/-}$ cells caused multipolar spindles similar to HeLa cells, where p53 is inactivated by the E6 protein from HPV. Expression of wild-type CENP-A, but not the methylation mutant of CENP-A rescues the phenotype observed in the HCT116 $p53^{-/-}$ cells (Fig. 5c,d). However, HCT116 cells with an intact p53 gene did not show the multipolar phenotype either under CENP-A suppressed conditions or when rescued with non-methylated CENP-A.

Consistent with defective centromeres, we observed an approximately two-fold increased rate of chromosome segregation defects in the CENP-A methylation mutants in the HCT116 cells (Fig. 5e–h). The increased degree of chromosome missegregation was observed in CENP-A methylation mutant both in $p53^{-/-}$ and $p53^{+/+}$ cells, when the endogenous CENP-A was knocked down. Therefore, although the rate of multipolar spindle defects observed under CENP-A suppression and rescue by the methylation defective CENP-A was dependent on the activity of p53, the defects in chromosome segregation was p53 independent (Fig. 5e–h). This suggests that p53 has a role in monitoring spindle defects but is not sensitive to chromosome missegregations.

To confirm the role of p53 in the loss of CENP-A methylation mediated spindle pole defects, we knocked down p53 in parental RPE or RPE cells where endogenous CENP-A was replaced with either the $^{WT}$CENP-A or $^{MT1}$CENP-A. The $^{MT1}$CENP-A replaced cells showed a significantly increased degree of multipolar spindles (Fig. 6a–d). These data suggest centromere/kinetochore defects cooperate with p53 loss to form chromosome instability. And that CENP-A methylation defects along with a reduction in p53 activity are sufficient to instigate multipolar spindles.

Since CENP-T and CENP-I were reduced in methylation mutant replaced cells, we tested whether siRNA against either

CENP-T or CENP-I caused multipolar spindle formation (Fig. 6e). We found that both CENP-T and CENP-I knockdown cause increased mitotic index demonstrating the effectiveness of the siRNAs (Fig. 6f). However, only CENP-T knockdown caused multipolar spindles (Fig. 6g,h). Similar to the CENP-A methylation mutant, the multipolar spindle formation is dependent on p53 status (Fig. 6i). Suggesting CENP-A methylation acts through CENP-T to stabilize the microtubule spindle.

**Force imbalance causes mitotic spindle multipolarity.** Force balance within the mitotic spindle generated by different motor

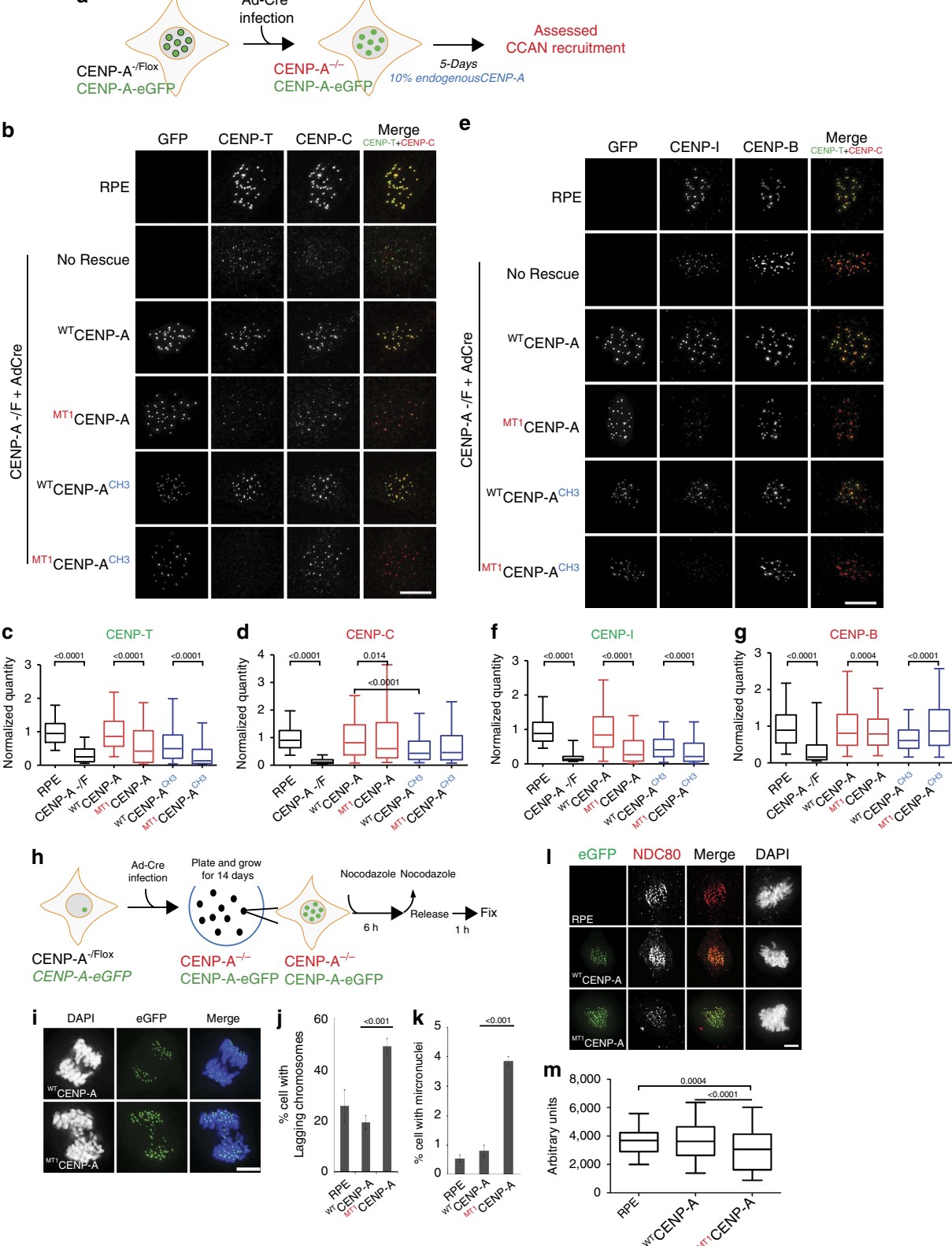

proteins such as Kif15, Eg5, CENP-E and Dynein is required for the formation of the bipolar spindle[30,32]. Alterations in the centromere and kinetochore structure can lead to improper kinetochore spindle attachment and altered dynamics of these factors. To determine whether the multipolarity is caused by imbalances in motor protein activity, we partially inhibited the motor protein Eg5 using very low concentration of Monastrol that did not cause monopolar spindle (Fig.7). We targeted Eg5 because suppression of dynein activity leads to multipolar spindles similar to what we observe when CENP-A is suppressed[30,32–34], and the phenotypes of dynein inhibition can be rescued by treatment with Eg5 inhibitor[32].

CENP-A was suppressed in HeLa cells by shRNA and cells were synchronized using thymidine, released and arrested in mitosis after 12 h of release by the addition of MG132 with or without a low dose of the Eg5 inhibitor Monastrol (Fig. 7a). The percentage of cells with multipolar spindles in the CENP-A shRNA treated cells was reduced to levels comparable to controls after treatment with Monastrol (Fig. 7b,c). This suggests that a force imbalance generated by the improper kinetochore formation after knockdown of CENP-A is the reason for the formation of multipolar spindle. To determine whether a similar force imbalance causes multipolar spindles in CENP-A methylation mutants, we rescued shRNA treated cells with either CENP-A wild type or methylation mutant CENP-A. We treated these cells with MG132 along with increasing concentration of Monastrol. With increase in Monastrol concentration we observed a decrease in multipolar spindles in CENP-A methylation mutant (Fig. 7d). This suggests that force imbalance causes pole fragmentation and multipolarity when CENP-A cannot be methylated. The origin of this force imbalance may be defective kinetochore function that promotes chromosome missegregation.

**Loss of CENP-A methylation confers a growth advantage**. Since loss of CENP-A methylation leads to mitotic defects, reduced fidelity of chromosome segregation and multipolar spindle defects, which are all hallmarks of cancer, we determined whether loss of CENP-A methylation provides any proliferative advantage. To address the proliferation and colony forming potential of cells harbouring non-methylated CENP-A we examined p53 wild-type and $p53^{-/-}$ HCT116 cells lines expressing normal or $^{MT1}$CENP-A, when endogenous CENP-A was suppressed by shRNA. As a population, 6 days after plating the mutant $^{MT1}$CENP-A expressing cells grew slightly faster than wild-type expressing cells, with doubling times of $20.5 \pm 0.2$ versus $23.1 \pm 0.9$ h, respectively (Supplementary Fig. 4E). Colony formation was assayed using the HCT116 $p53^{+/+}$ and $p53^{-/-}$ cells after knockdown and replacement of endogenous CENP-A with wild-type or CENP-A methylation mutant ($^{MT1}$CENP-A) (Fig. 8a). Cells expressing $^{MT1}$CENP-A formed larger colonies in

a $p53^{-/-}$ background, when compared to CENP-A wild type cells after 8 days of endogenous CENP-A depletion by shRNA and plating 500 cells into 10 cm plate under continued expression of the shRNA for 14 days (Fig. 8b,c). Single cells were sorted into 96-wells and individual cells were assessed for their growth after 16 days in culture (Fig. 8e). While most $^{MT1}$CENP-A expressing cells grew similarly to wild-type CENP-A expressing cells, a subset of $^{MT1}$CENP-A cells grew more rapidly. Suggesting a clonal evolution of some mutant expressing cells (Fig. 8b–e). These data suggest that in the absence of p53, the loss of CENP-A methylation provides a proliferative advantage and thus may promote tumorigenesis. After endogenous CENP-A shRNA suppression for 8 days we injected these cells into nude mice and looked for the appearance of measurable tumours under continued induction of shRNA (Fig. 8a). $p53^{-/-}$ cells formed tumours in all animals; however, first appearance of measurable tumours was significantly earlier when cells expressed the CENP-A methylation mutant compared to cells expressing wild-type CENP-A or parental HCT116 $p53^{-/-}$ cells (Fig. 8f). The $^{MT1}$CENP-A or $^{WT}$CENP-A overexpression was not sufficient to increase the rate or prevalence of tumour formation when $p53^{+/+}$ was normal in these cells (Fig. 8g).

## Discussion

The ability of the CENP-A nucleosome to recruit a unique set of chromatin-associated factors is key to its ability to delineate centromeric chromatin from canonical H3 containing regions. Here we show that the α-amino trimethylation contributes to the ability of CENP-A to recruit essential centromeric components (Fig. 8h). The α-amino trimethylation of several proteins has been demonstrated, but in most cases the functional implication of this modification remains poorly understood[23–26,28]. Two lines of evidence show that α-amino trimethylation is essential for CENP-A amino terminal tail function. First we demonstrate that under conditions where cell viability is dependent on the CENP-A amino terminus (that is, in CENP-A knockout RPE-1 cells expressing CENP-A$^{CH3}$), that elimination of CENP-A methylation renders the cell inviable. Second, we show that the phenotype of suppressing CENP-A by shRNA cannot be rescued by non-methylated CENP-A. Deregulation of the CENP-A α-amino trimethylation causes chromosome segregation defects such as lagging chromosomes and multipolar spindle formation.

Multiple interactions between the CENP-A nucleosome and the CCAN contribute to the assembly of the CCAN at centromeres, and CENP-A N-terminal and C-terminal tails have partially redundant function in recruiting the CCAN. Direct and indirect interactions have been identified between the CENP-A amino and carboxyl termini and CENP-C. Consistent with this redundancy, the N-terminal and C-terminal tails along with the centromere targeting domain of CENP-A together are required

**Figure 3 | CENP-A α-amino methylation is required for CCAN formation.** (**a**) Schematic of the experiment. (**b**) Representative images of the cells immunostained for CENP-T and CENP-C. Centromeric CENP-T was decreased upon CENP-A knockout and is rescued by replacement of the wild type CENP-A but not with methylation mutant. (**c,d**) Quantitation of CENP-T and CENP-C at the centromere after removal of endogenous CENP-A by cre-recombinase. $n = 2$. A minimum 30 cells in two biological replicates were used for the quantitation. (**e**) Representative images of cells immunostained for CENP-I and CENP-B. CENP-I level decreased when endogenous CENP-A was removed by Cre expression and is rescued by replacement with the wild type CENP-A but not with methylation mutant. (**f,g**) Quantitation of CENP-I and CENP-B at the centromere after removing endogenous CENP-A by cre-recombinase. Box-and-whisker plots. Central lines, medians; whiskers, range 5–95 percentile; outliers not shown. (**h**) Schematic of the experiment in I-K. eGFP tagged CENP-A wild type and mutant integrated RPE-CENP-A -/F cells were infected with cre-recombinase virus to remove endogenous CENP-A. Individual colonies were picked after 14 days and expanded. The cells were treated with 0.1 μM Nocodazole for 6 h and then released for one hour, fixed and DAPI stained (**i**) Scale bar, 10 μm. Cells expressing the methylation mutant showed significantly high percentage of (**j**) lagging chromosomes and (**k**) micronuclei formation in the absence of Nocodazole treatment. The experiments were done three times. Mean and s.d. shown. (**l**) Representative images of the RPE cells immunostained for NDC80. Kinetochore localization of NDC80 was decreased in methylation mutant CENP-A. (**m**) Quantitation of NDC80 at the kinetochore after removal of endogenous CENP-A by cre-recombinase and replacement of either wild type or mutant CENP-A. A minimum of 8 prometaphse cells were used for the quantitation. Indicated $P$ values were determined by $t$-test. Scale bars, 10 μm.

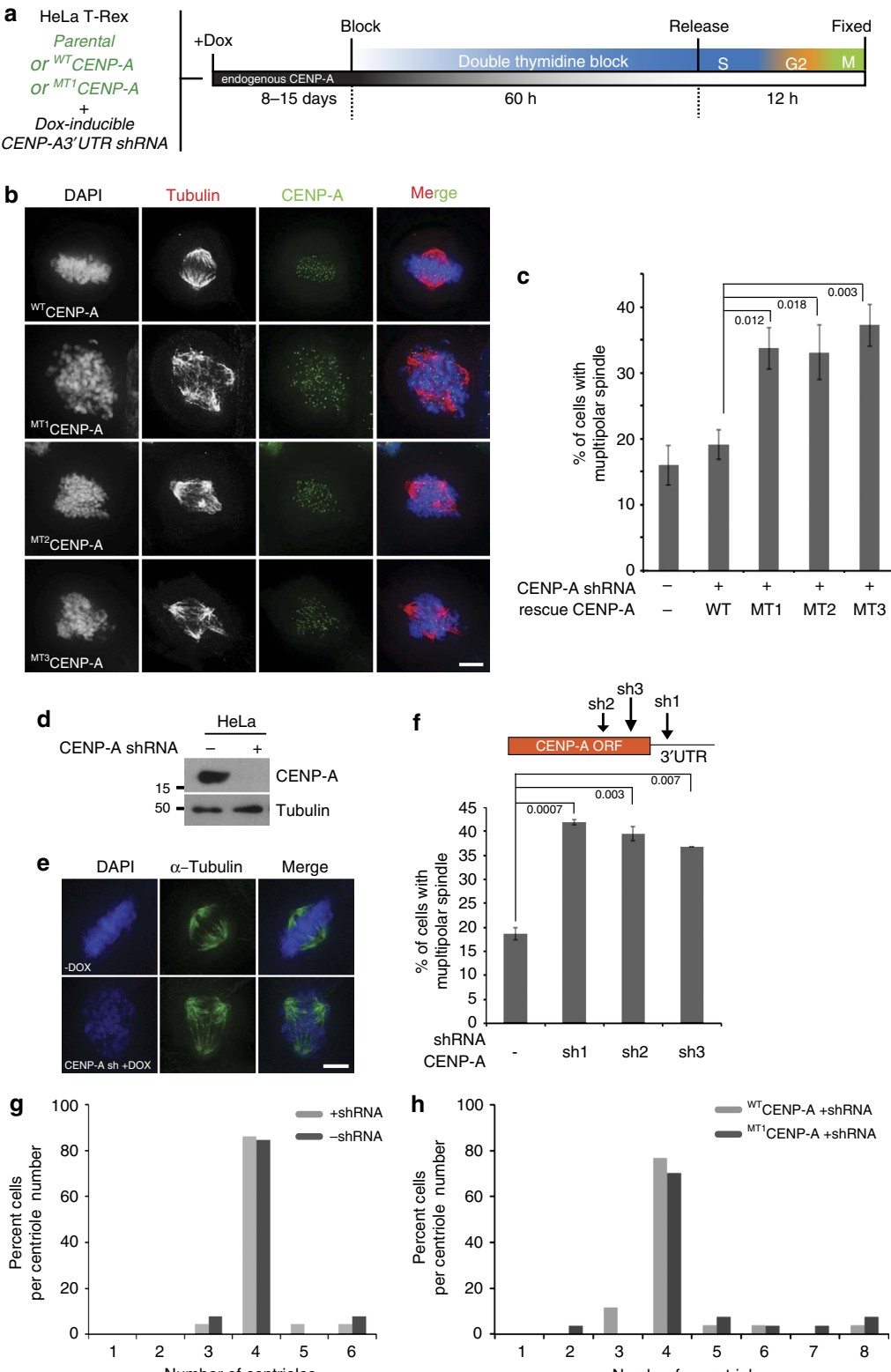

**Figure 4 | CENP-A α-amino methylation required for maintaining bipolar spindle.** (**a**) Schematic of the CENP-A shRNA treatment of parental and CENP-A-eGFP expressing HeLa cells. (**b**) Rescue with methylation deficient mutants of CENP-A fails to rescue multipolar spindle defects. (**c**) Quantitation of the multipolar cells. $n = 3$. Error bars indicate s.d. Indicated $P$ values were determined by $\chi^2$ test. (**d**) Western blot showing reduction of CENP-A protein following shRNA treatment. (**e**) CENP-A knockdown caused cells to develop multipolar spindles. Cells were stained for α-tubulin after CENP-A suppression by shRNA. (**f**) Quantitation of multipolar cells following CENP-A shRNA treatment. $n = 3$. Error bars indicate s.d. Indicated $P$ values were determined by $\chi^2$ test. (**g,h**) Number of centrioles in the multipolar cells formed by knockdown and replacement with the wild type and mutant CENP-A. The majority of cells have four centrioles, consistent with normal cytokinesis and centriole duplication. $n > 25$ mitotic cells. Scale bars, 10 μm.

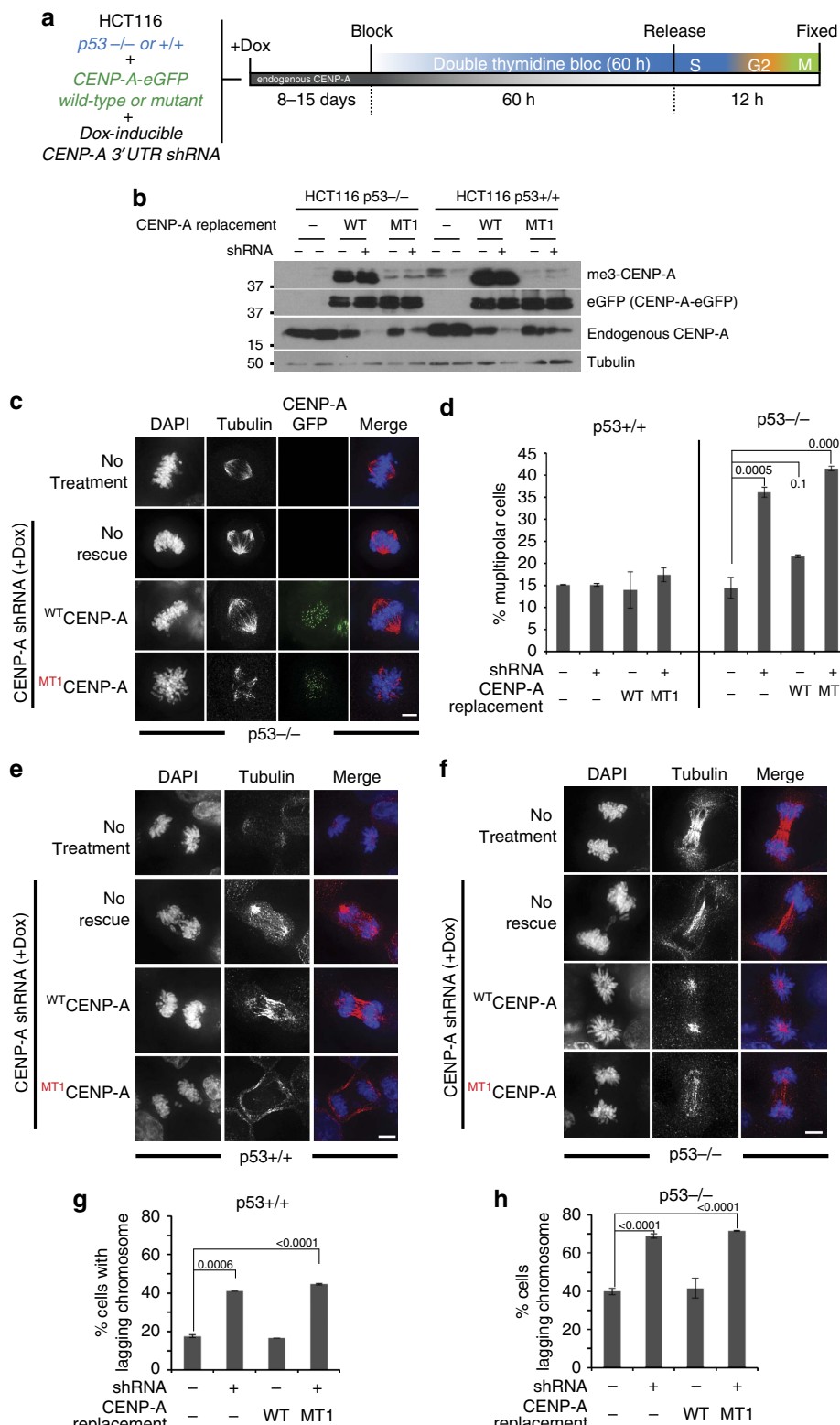

**Figure 5 | CENP-A α-amino methylation required for maintaining chromosome segregation fidelity. (a)** Schematic of the CENP-A shRNA experiment. **(b)** Western blot showing endogenous and tagged CENP-A levels following shRNA treatment. **(c)** CENP-A shRNA mediated suppression causes multipolar spindles in p53$^{-/-}$ cells, but not in p53$^{+/+}$ HCT116 cells. Cells were stained for α-tubulin. **(d)** Quantitation of the multipolar cells. **(e,f)** Wild-type and p53 null cells both showed increased lagging chromosomes after CENP-A shRNA mediated suppression. Replacement with methylation mutant CENP-A in p53$^{+/+}$ and p53$^{-/-}$ HCT116 failed to rescue the chromosome segregation defects. Scale bars, 10 μm **(g,h)** Percentage of cells showing lagging chromosomes after CENP-A knockdown and replacement in the indicated cell lines. The experiments were done in duplicate. Error bars indicate s.d. Indicated P-values were determined by $\chi^2$ test.

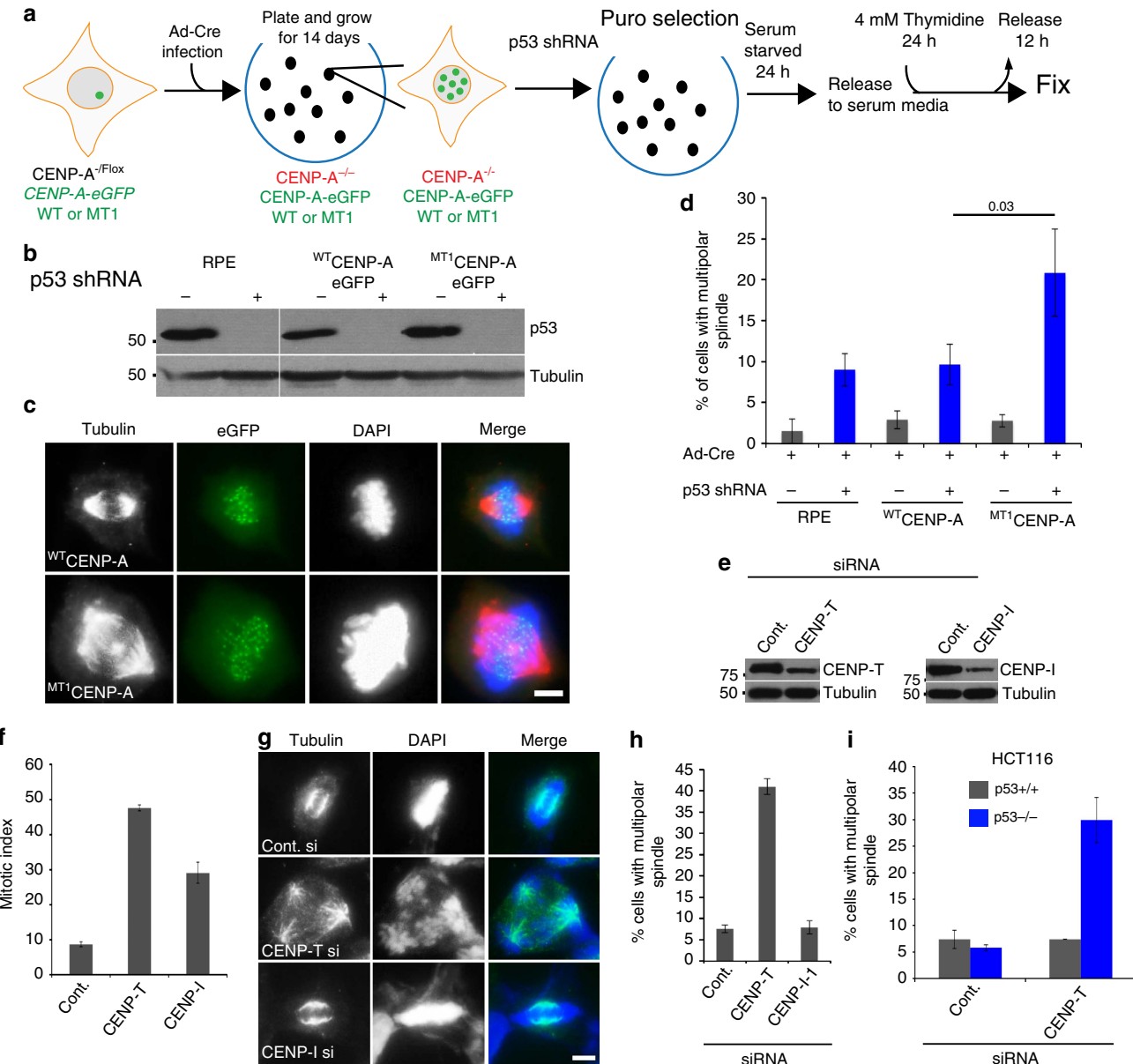

**Figure 6 | Loss of CENP-T causes multipolar spindle formation in a p53 dependent fashion.** (**a**) Schematic of the experiment in B-D. eGFP tagged CENP-A wild type and mutant integrated RPE-CENP-A -/F cells were infected with cre-recombinase virus to remove endogenous CENP-A. Individual colonies were picked after 14 days and cells were infected with p53 shRNA. The p53 shRNA integrated clones were selected with puromycin. The cells were then synchronized and fixed. (**b**) Western blot showing the efficiency of p53 knockdown. (**c**) Cells showing multipolar spindles in MT1CENP-A replaced cells but not in wild-type. (**d**) Percentage of cells showing multipolar cells in wild type and mutant cells, based on three independent experiments. (**e**) Western blots showing siRNA mediated knockdown of CENP-T and CENP-I in HeLa T-REx cells. (**f**) Loss of CENP-T and CENP-I leads to mitotic arrest, based on three independent experiments. (**g,h**) CENP-T knockdown but not CENP-I cause multipolar spindle in HeLa T-Rex cells, based on three independent experiments. (**i**) CENP-T knockdown causes multipolar spindles only in p53 null HCT116 cells but not in p53 wild-type HCT116 cells, based on three independent experiments. Error bars indicate s.d. Indicated *P* value was determined by $\chi^2$ test. Scale bars, 10 μm.

for cell survival[14]. We show that an essential function of the CENP-A amino terminus depends on its methylation. CENP-T in humans, and CENP-T and CENP-I in S. *pombe*, requires the CENP-A amino terminus for accumulation at centromeres[4,16]. On the basis of a recent report[35], this may reflect a selective effect of CENP-A methyation on CENP-T localization and stability. Although, we do not know whether S. *Pombe* CENP-A is methylated, the amino terminal sequences of several vertebrate CENP-A proteins are *in vitro* substrates for NRMT1, suggesting that CENP-A methylation is a conserved feature of the protein[28]. We show that CENP-I and CENP-T levels are reduced from

the centromere when CENP-A is unmethylated, showing that methylation is a key feature of the human CENP-A amino terminus that is required for CENP-T and CENP-I localization. Consistent with the reduction of these proteins, the kinetochore localization of NDC80 was also reduced. Dominant negative forms of NDC80, and NDC80 disruption in some cell types, can lead to multipolar spindles through a similar mechanism of centriole disengagement[36,37]. Knockdown of CENP-T also leads to multipolar spindle formation, suggesting that the loss of CENP-T may be the reason for the effect of CENP-A methylation mutants on spindle integrity.

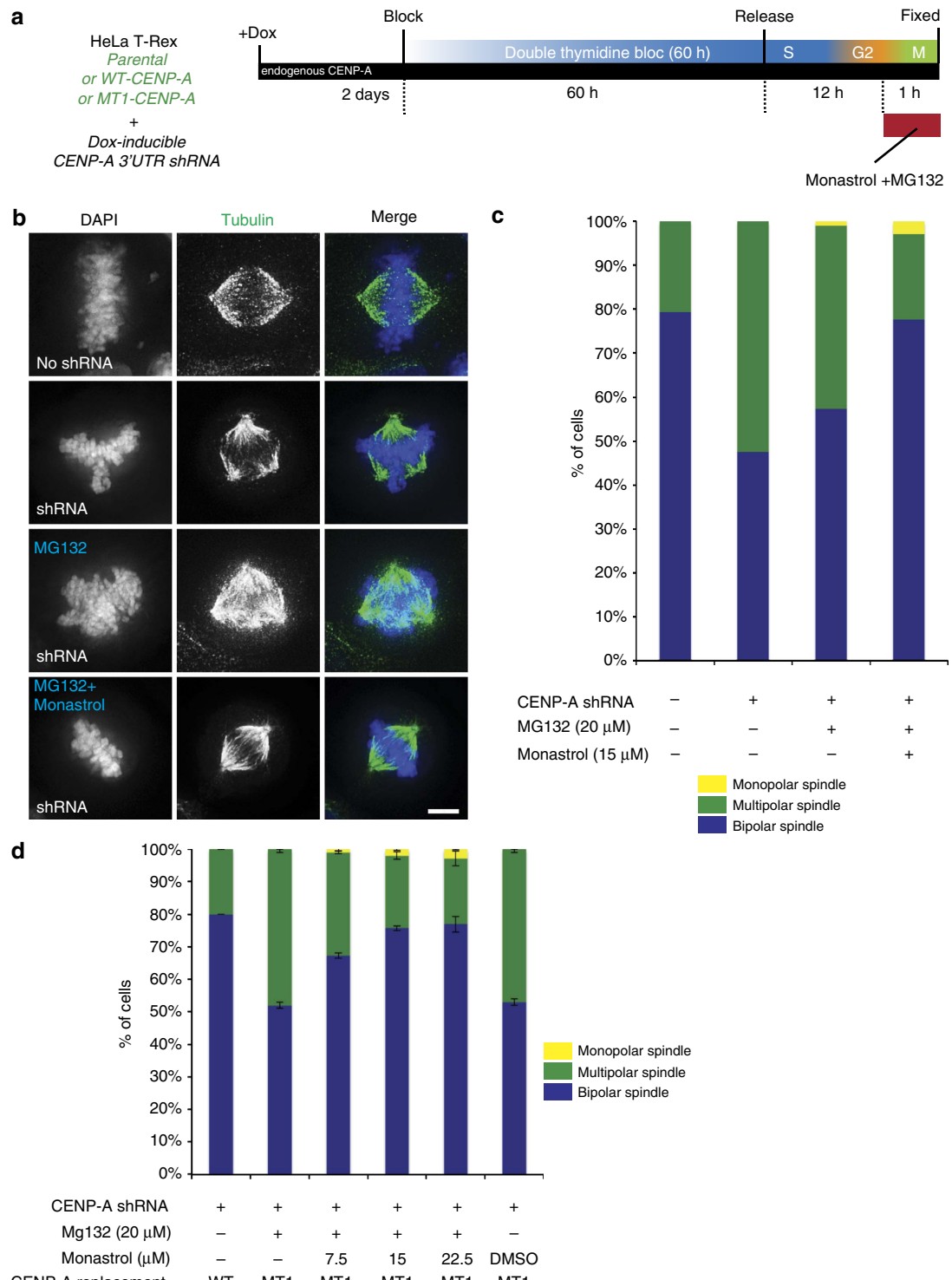

**Figure 7 | Eg5 inhibition rescue multipolarity caused by loss of CENP-A methylation.** (**a**) Schematic of the experiment. CENP-A was knocked down for two days prior to conducting a double thymidine block and release. Twelve hours after the second release, cells were treated with Mg132 or Mg132 and Monastrol for one hour. (**b**) Immunofluorescence analysis showing multipolar spindles formed in cells treated with CENP-A shRNA are rescued by treating cells with the Eg5 inhibitor Monastrol. Scale bar, 10 μm. (**c**) Graph showing fractions of monopolar, bipolar and multipolar cells in cells treated with Eg5 inhibitor after CENP-A shRNA treatment. (**d**) Graph showing fractions of monopolar, bipolar and multipolar cells in cells treated with various concentrations of Eg5 inhibitor after CENP-A shRNA treatment and replacement with CENP-A wild-type and methylation mutant. The experiment was done in duplicate and ∼100 mitotic cells were counted in each condition. Error bars indicate s.d. Scale bars, 10 μm.

We show that NRMT1 is the enzyme responsible for CENP-A trimethylation *in vivo*. We made several mutants that either eliminate CENP-A methylation by disrupting the recognition sequences within CENP-A (MT2 and MT3) or by adding an additional alanine to the amino terminus (MT1), which we predict will misplace the amino terminus from the active site of NRMT1 (refs 38,39). Mutations that preclude CENP-A methylation phenocopy the effect of NRMT1 shRNA

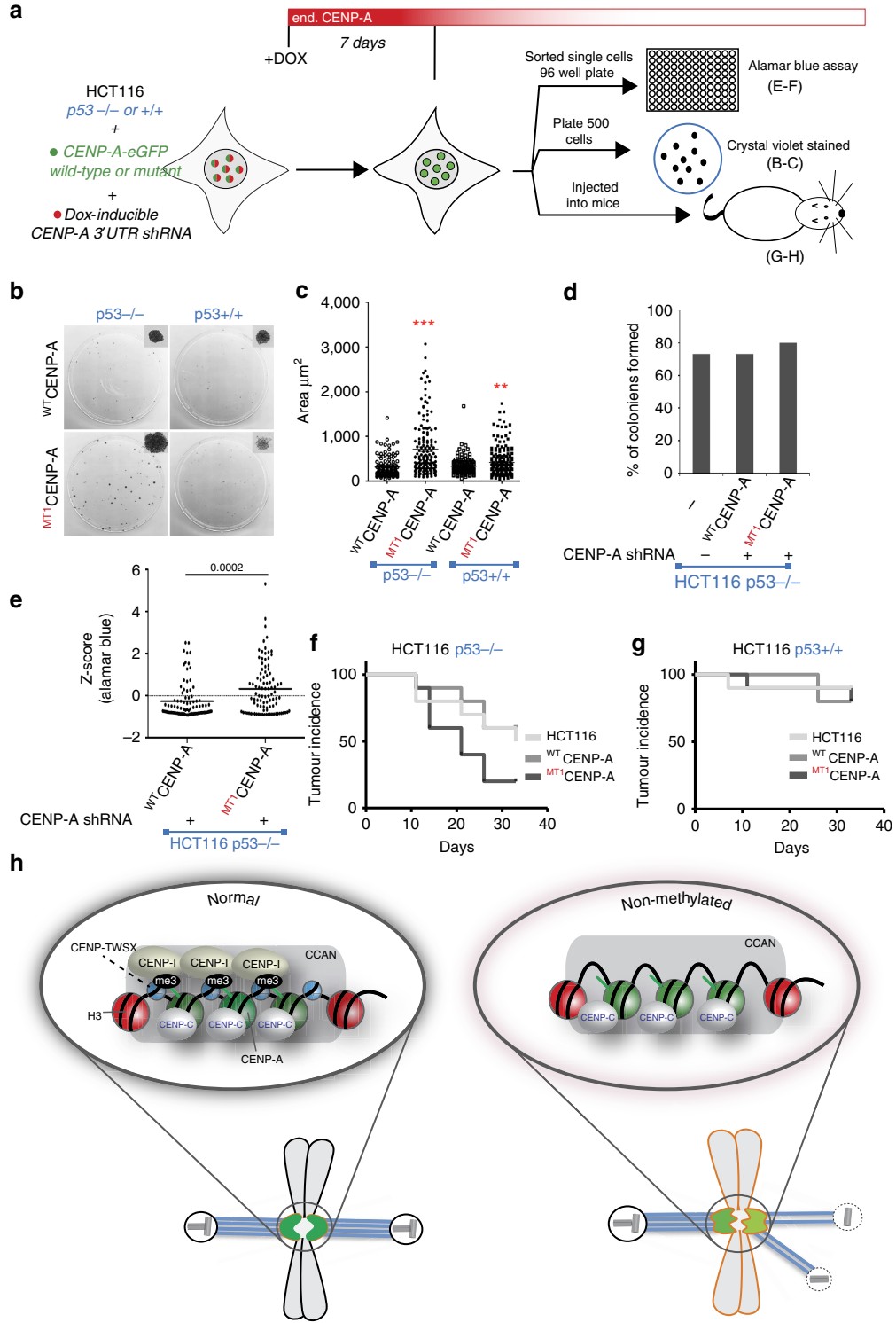

**Figure 8 | Loss of CENP-A α-amino methylation increases survival of the HCT116 cancer cells.** (**a**) Schematic of the experiments in this figure using HCT116 cells. Endogenous CENP-A knockdown in CENP-A-eGFP wild type and mutant expressing cells. (**b**) Crystal violet stained cell colonies. (**c**) Quantitation of the colony size observed in each condition. The experiment was done in triplicates. P values, \*\*\*P value < 0.00001, \*\*P value < 0.01 as determined by t-test. (**d**) Number of colonies formed on a 96 well plate after sorting single cells into each well. There was no significant difference between parental and CENP-A wild-type or mutant1 replaced HCT116 p53$^{-/-}$ cells. (**e**) Clone growth measured by the Alamar blue assay showing some HCT116 p53$^{-/-}$ cells replaced with $^{MT1}$CENP-A form faster growing colonies. 100 cells were plated for the assay in individual wells. Indicated P value were determined by t-test. (**f,g**) Kaplan–Meier curve showing time to measurable tumour appearance (tumour incidence) in mice bilaterally injected with either CENP-A wild type or methylation mutant cells in five mice. (**f**) p53$^{-/-}$ ($^{WT}$CENP-A versus $^{MT1}$CENP-A P value = 0.05) cells and (**g**) p53$^{+/+}$ cells. (**h**) Model showing the function of the α-amino terminal methylation of CENP-A. CENP-A methylation is required for the recruitment of CCAN proteins into the centromere and loss of methylation leads to multipolar spindle formation.

suppression[23]. CENP-A methylation mutants are efficiently recruited to centromeres; therefore, methylation is not required for deposition at the centromere. The dynamics of CENP-A methylation, which peaks in mitosis, may explain in part the pattern of CENP-T recruitment to the centromere, which gradually increases beginning in S-phase through G2 and into mitosis[40].

The response of cells to a reduction of CENP-A differed depending on the p53 status of the cells. We observed phenotypic differences when comparing of CENP-A suppression and methylation in RPE-1, which are an untransformed mostly diploid cell line expressing active p53, and HeLa cells, in which p53 is inactivated by the HPV E6 protein; and again when we compared HCT116 cells that differ only in the presence of the p53 gene, or RPE cells where p53 was suppressed by shRNA. Interestingly, chromosome segregation defects were observed as a result of CENP-A suppression or expression of the CENP-A methylation mutant regardless of p53 status. In contrast, we observed the presence of multipolar spindles in response to CENP-A suppression or mutant expression only when p53 was inactive. We demonstrated that altering the forces applied to the spindle by inhibiting Eg5 rescues the effect of CENP-A depletion or methylation mutants on spindle integrity. Reduced centrosome cohesion in p53 negative cells may render the centrioles more prone to multipolar spindle formation by pole fragmentation[41,42]. Alternatively, the role of p53 in centrosome surveillance may eliminate these cells from the population[43].

Overexpression of CENP-A and its chaperone HJURP has been observed in a variety of cancers[44–46]. Recently, the reduction of NRMT1 was associated with increased tumour size in a breast cancer xenograft model[29]. There is no indication that NRMT1 levels are co-regulated with CENP-A; therefore, with increased expression of CENP-A, the proportion of methylated CENP-A at the centromere may be less, possibly leading to increased formation of multipolar spindles and chromosome missegregation. Recent work has demonstrated that the missegregation of chromosomes and the formation of micronuclei is sufficient to induce genomic rearrangement of the missegregated chromosome, a process known as chromothripsis[47]. These events are limited in the presence of wild-type p53. Our results show that CENP-A overexpression and CENP-A methylation mutants are not sufficient to cause tumour formation; however, in a p53 negative background loss of CENP-A function promotes tumour formation. This study, thus, uncovered a novel mechanism, where centromere cooperates with p53 to prevent chromosome instability.

## Methods

**Cell lines and plasmids.** HCT1116 p53$^{-/-}$ and HCT116 p53$^{+/+}$ cells were grown in DMEM with 10% fetal bovine serum (FBS) (ref. 31). HeLa Flp-In cells (Gift from Stephen Taylor) were used to make stable lines, and were grown in DMEM with 10% FBS and hygromycin. Both RPE-1 and RPE-1 CENP-A -/F (ref. 14) cells (gift from D. Cleveland) were grown in DMEM F12 medium with 10% FBS. A modified version of the LAP tag was used to make CENP-A-eGFP on pCDNA5/FRT destination vector, where the destination cassette was put in frame with eGFP followed by S-tag[48] (kindly provided by Todd Stukenberg lab).

**In vitro methylation assays.** The in vitro methylation assay was performed using recombinant NRMT1 (ref. 23). 6XHis tagged Human NRMT1 (Gift from Ian Macara) was purified from Escherichia coli and used for the methylation assays. CENP-A amino acids 2–10 were purified, as a fusion protein with GFP. 6XHis followed by a Factor X cleavage site was placed N-terminal to Gly1 of CENP-A. Recombinant proteins were expressed and purified in BL21 E. coli using Ni-NTA beads. The 6XHis tag was removed by cleavage with Factor X (Sigma-Aldrich). The cleaved proteins were negatively selected with Ni-NTA column and the eluate was used as substrate for methylation assays. For the methylation assay, 3.0 pmol of recombinant NRMT + 1 µg of purified substrates were incubated in MTase buffer (50 mM Tris, 50 mM potassium acetate, pH 8.0) for 2 h at 30 °C after addition of 1 µl of $^3$H-SAM (0.55 µCi µl$^{-1}$). The proteins then bound to nitrocellulose

membrane by filtering. The membrane was washed with 50 mM sodium bicarbonate and the radioactivity was measured using scintillation counter. For western blotting analysis, cold SAM was used instead of $^3$H-SAM.

**Virus production and shRNA treatment.** The human and mouse shRNA pGIPZ constructs containing shRNA against human NRMT (AGAGAAGCAATTCTAT TCCAAG), and mouse NRMT (CCCTGCCAGACAGTACCAATTA, used as control) were obtained from Ian Macara's lab[23]. CENP-A targeted pTRIPZ shRNA plasmids RHS4696-100901553 (3′UTR, AGACTGACAGAAACACTGG), RHS4696-100902676 (CDS, TTGGGAAGAGAGTAACTCG), RHS4696-99703417 (CDS, TGAACTAGAAATGCTTCTG) were obtained from Dharmacon. p53 shRNA was provided by K. Oegema (UCSD). The viruses were made in 293LT cells. The cells were co-tansfected with NRMT pGIPZ or CENP-A pTRIPZ shRNA along with vesicular stomatitis virus coat protein plasmid (pMD2G), and packaging plasmid (psPAX2). Viral supernatants were collected after 48 h and infected. Puromycin (2 µg ml$^{-1}$) was added to HeLa T-Rex or HCT116 cells after 3 days to select transduced cells. For CENP-A knockdown, shRNA was induced two days before starting a double thymidine block and release.

For knockdown and replacement experiments in HCT116 cells, C-terminally eGFP-tagged CENP-A wild type and mutants were cloned into pBABE-Bla retrovirus vector by using cold fusion (System Biosciences,). Pseudotyped MULV viruses were packaged in HEK 293 GP cells along with VSVG plasmid. The viruses were collected after 3 days. For making double stable integration, HCT116 cells were co-infected with CENP-A retrovirus and CENP-A 3′UTR lentivirus. The double stable cells were selected using 6 µg ml$^{-1}$ Blasticidin and 2 µg ml$^{-1}$ puromycin. CENP-A was also cloned into a modified pcDNA5/FRT-LAP destination vector kindly provided by Stukenberg lab using Gateway system (Invitrogen). HeLa T-Rex Flp-In cells were transfected with CENP-A wild type or mutants along with FLP-recombinase and stable integrations were selected with hygromycin (200 µg ml$^{-1}$). The stables cells were then infected with doxycycline inducible CENP-A shRNA lentivirus to establish double stable cell lines. For knockdown replacement experiments shRNA was induced 8 days before starting double thymidine block and release to obtain effective endogenous CENP-A depletion. The CENP-T was knockdown using siRNA 5′-AGAAGUGCCUAGA UAAAUA-3′ (Life technologies) and CENP-I using smart pool siRNA (Thermo Fisher Scientific, M-029617-01). Cells were treated with 20 nM siRNA for 40 h.

**Colony formation assay.** RPE CENP-A -/F cells, $4 \times 10^4$ cells were plated in a 12-well plate. The cells were washed three times in DMEM:F12 medium containing 2% FBS the following day and infected with Ad-Cre virus in 400 µl of DMEM:F12 medium containing 2% FBS. The cells were washed three times with DMEM:F12 medium containing 10% FBS after 4 h. Cells were grown 2 more days, and then 500 cells were plated in triplicate on a 10 cm$^2$ dish. Colonies were fixed after 14 days for 10 min in methanol and stained for 10 min using a crystal violet staining solution (0.05% crystal violet and 25% methanol). Clonogenic survival was calculated by dividing the number of colonies formed in the Ad-Cre-treated condition versus the untreated cells. For the HCT116 cells, endogenous CENP-A was knockdown using doxycycline (1 µg ml$^{-1}$) inducible 3′UTR shRNA for 7 days. Five hundred cells were then plated in triplicate on a 10 cm$^2$ dish in the presence of doxycycline. Colonies were then fixed after 14 days.

**Drug treatment.** Cells were double thymidine blocked and released. After 12 h of release, the cells were either treated with DMSO as a control or monastrol in the presence of Mg132. Mg132 alone was added as a control. One hour after drug treatment, cells were fixed in 4% formaldehyde and stained for tubulin.

**Immunofluorescence.** Cells were either pre-extracted with 0.1% triton in PHEM buffer (60 mM PIPES, 25 mM HEPES, 10 mM MgCl$_2$) or PBS for 3 min and then fixed in 2–4% formaldehyde for 10 min or fixed in 2–4% formaldehyde for 10 min and then permeabilized in 0.1% TritonX-100 for 5 min. For staining γ-tubulin and centrin, cells were fixed in ice-cold methanol on ice for 3 min, and then proceeded with antigen blocking. The antibodies used for immunofluorescence were: mouse anti-CENP-A (1:1,000, Cat#13939 Abcam), rabbit anti-me3-CENP-A (1:200), mouse anti-CENP-B (1:250), rabbit anti-CENP-T (1:2,000), rabbit anti-CENP-I (1:1,000), mouse anti-CENP-C (1:1,000), rabbit anti-γ-tubulin (1:1,000; Cat# T5192 Sigma), mouse anti-centrin (1:1,000; Cat# 04-1624 Millipore), mouse anti-α-tubulin (1:1,000). The cells were then blocked in 2% BSA and 2% fetal calf serum with 0.1% triton in PBS and incubated with primary antibodies 1 h and then secondary antibody was added for 1 h. All washes between each step were done using PBS + 0.1% TritonX-100. DNA was stained with DAPI (0.2 µg ml$^{-1}$) and cells were mounted in Prolong (P10144, Thermo fisher). Images were acquired using a Deltavision optical sectioning deconvolution instrument (Applied Precision) using a × 60 or × 100 oil-immersion Olympus objective lens connected with Photometrics CoolSNAP HQ$^2$ camera and softWoRx acquisition software. Images were deconvolved and presented as maximum stacked images[49]. The quantitation of the centromere localization of the CCAN proteins was performed by using ImageJ and CRaQ (ref. 50). Images were also acquired using Spinning Disk Confocal microscope and quantitated using Volocity 5.5. Unpaired t-tests were performed to see the difference between groups.

**Mouse xenografts experiments.** All xenograft experiments were done in Molecular Assessments & Pre-Clinical Studies (MAPS) core, University of Virginia according to the guidelines of the institutional review board and ethics committee. HCT116 cells doubly integrated with CENP-A construct and CENP-A 3′UTR shRNA were treated with Dox ($1 \mu g\, ml^{-1}$) for 8 days to knockdown endogenous CENP-A. $2.5 \times 10^5$ cells were then bilaterally injected subcutaneously. For the continuous expression of the CENP-A shRNA, the Dox was added in the drinking water. The tumour volume was measured twice a week for thirty days. A total of five mice were used in each subgroup.

**AlamarBlue cell viability assay.** After 10 days of treatment with doxycycline to induce CENP-A 3′UTR shRNA in CENP-A wild-type or mutant stable expressing HCT116 $p53^{-/-}$ cells, single cells were sorted into 96 well plate. The cells were grown 16 days with doxycycline. AlamaeBlue reagent (Biorad) added to the culture media (10% of the media, $10 \mu l$ reagent to $100 \mu l$ media). The cells were incubated 4 h in the tissue culture incubator and the fluorescence was measured at 590 nm. Additional methods are available in Supplementary section.

**Data availability.** All data generated or analysed during this study are included in this published article or in Supplementary information files.

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

## Acknowledgements

We would like to thank Don W. Cleveland, Ian G. Macara, Christine Tooley, Janusz J. Petkowski and Karen Oegema for reagents. We thanks to the Todd Stukenberg and Dan Burke labs for sharing reagents and comments. K.M.S. was supported by Department of Defense Visionary Postdoctoral Fellowship (W81XWH-13-1-0106). D.R.F. was supported by NIH R01GM111907 and by a Research Scholar Award from the American Cancer Society.

## Author contributions

K.M.S. and D.R.F. conceived and designed the experiments. K.M.S. executed the experiments. K.M.S. and D.R.F. wrote the manuscript. D.F. developed the RPE-CENP-A$^{-/F}$ cell line, performed experiment in S3G and contributed to editing the manuscript.

## Additional information

**Competing financial interests**: The authors declare no competing financial interests.

**How to cite this article**: Sathyan, K. M. *et al.* α-amino trimethylation of CENP-A by NRMT is required for full recruitment of the centromere. *Nat. Commun.* **8**, 14678 doi: 10.1038/ncomms14678 (2017).

**Publisher's note**: 

