## [Peer Review file · Nature Communications]

Reviewers' comments:

Reviewer #1 (Remarks to the Author):

Though this laboratory has previously shown that CENP-A is N-terminally methylated by the methyltransferase NRMT1 in vitro, they now follow up this work and verify that the same occurs in vivo. They also identify novel functional roles of this PTM, including promoting survival and recruitment of other centromere proteins and preventing lagging chromosomes and spindle pole defects.

This work is convincing and will drive the field forward, as there is currently a great need for categorization of the effects of N-terminal methylation on the function of individual substrates. I believe it is suitable for publication in Nature Communications, if the following points are addressed.

- 1.) In both the introduction and the discussion, it is stated that RCC1 is the only NRMT1 substrate for which the impact of N-terminal methylation is known. This is untrue, and authors should include papers showing the impact of N-terminal methylation on DDB2 and CENP-B function (Cai et al, JBC, 2014; Dai et al., Journal of Proteasome Research, 2013).
- 2.) Authors state that the CENP-A mutants MT1 and MT2 are recognized by an antibody against phospho-Ser16/18 (Figure 1J). However, MT2 is not included in this figure, and the Ser16/18 phospho status of MT3 is not mentioned. These data should be included.
- 3.) Figures 3B and D are meant to demonstrate a "significant" reduction of CENP-C levels in cells rescued with WTCENP-ACH3. However, this does not appear to be the case by eye (Figure 3B) and there is no denotation of significance in the quantification (Figure 3D). If significance is to be claimed, the proper statistics will have to be shown and explained. Authors also claim a "significant" reduction in CENP-T and CENP-I (Figures 3C and F) without use of statistics. This also needs to be rectified.
- 4.) It has long been hypothesized that N-terminal methylation could regulate protein-protein interactions, but there is little experimental proof. While Figure 3 hints at this, the impact of this paper could be significantly enhanced if the authors could directly demonstrate that unmethylated recombinant CENP-A (or mutant) binds less CENP-T or CENP-I.
- 5.) Figures 7D and E are labeled tumor free survival. As this term is usually meant to denote the length of time after primary treatment that an animal goes without signs of recurrence, it does not apply here. It is unclear if authors are measuring survival after injection or number of days until tumor appearance, but graphs should be relabeled to make this clear. The Materials and Methods section state tumor volumes were measured, this data should also be included, as the premise is that loss of methylation increases growth. The legend for Figure 7 does not match the figure.
- 6.) Some typos in figure legends. Last bar of Figure 2D is labeled CENP-A-/-, where everywhere else it is CENPA-/F. Supplemental Figure 2 refers to Ser7, while the text refers to it as Ser6. Supplemental Figures 3A and D are labeled + AdCre but are supposed to contain endogenous CENP-A.

Reviewer #2 (Remarks to the Author):

This is a beautiful study, very carefully executed and particularly the results of the first half of the story are very clear. The authors have cleverly worked around redundancies in the recruitment of kinetochore components to pinpoint the exact contribution of amino-terminal methylation of CENP-A. The first part of the story, which is centered around the methylation event itself and the consequences of impaired methylation is particularly strong, and I only have minor points on this part (see below).

The second part of the story has some gaps that have to be filled to complete the story. Since there currently is no evidence for context-dependent regulation of CENP-A methylation (other than

the cell cycle-dependent increase in methylation), the authors have focused the second part of the story on possible consequences of altered methylation in tumorigenesis. Again the experiments for this part are well designed and the outcomes are very interesting. The authors show effects on spindle assembly, and show that altered methylation affects tumor growth. However, the underlying basis for the observations is not always clear or investigated, and the manuscript becomes rather speculative. For example, the differences in spindle defects that are observed between the p53-proficient and deficient cells are interesting (and based on solid work), but the basis for this difference is unclear. Also, the effect on increased tumor growth is interesting, but if this is directly linked to the mitotic defects that are reported in the first part of the paper remains speculative. In my opinion, the authors are better off focusing on the spindle defects or the effects on tumorigenesis and close the indicated gaps, rather than including both, this would be more than sufficient to warrant publication.

Major points:

1. The authors suggest that the defects in chromosome alignment and spindle integrity are due to reduced recruitment of the CCAN complex. Most notably, they see reduction in recruitment of CENP-T and CENP-I. To nail this issue, they should show that reduction in CENP-I and/or -T levels, independent of alterations in CENP-A methylation cause a similar phenotype, and moreover, show that this phenotype can be rescued if CENP-I or -T is recruited in a methylation-independent manner. Admittedly, the latter experiment might not be possible if the reduction in CENP-I and -T occur independent from each other.
2. The authors very nicely show that the multipolar spindle phenotype that is observed upon perturbation of CENP-A methylation is due to a force imbalance. They fail to provide an explanation for this imbalance, and also fail to provide an explanation for the very interesting observation that the force imbalance is not apparent (or not causing a phenotype) in p53-proficient cells. I assume that the force imbalance is due to a reduction in K-fiber strength, but the fact that segregation errors are equally frequent in p53-proficient and -deficient cells suggests that K-fibers are affected in both settings? Does weakening kinetochore function in the same p53-proficient and deficient cells result in a similar difference?
3. The authors very nicely show that the growth of tumor cell lines is enhanced in tissue culture and in mice, when CENP-A methylation and p53-function are perturbed. But it remains speculative if this is directly linked to the mitotic defects described in the first part of the paper. The authors should discriminate if proliferation rates are affected, or if the enhanced size of colonies and tumor is due to enhanced survival of the cells under these conditions. Given that they report an increase in segregation errors, one would expect that effects on tumor growth are not a direct effect on proliferation, but rather caused by the evolution of more aggressive cell clones (than could of course proliferate faster). Do the colonies/tumors display heterogeneity in chromosome numbers, as could be expected? Does partial inhibition of Eg5 affect colony size in the p53-deficient cells (and not in the p53-proficient cells)?

Minor points:

Fig.1; blots are consistently labeled with me3-CENP-A, but shouldn't this be me3-CENP-A-GFP? Me3-CENP-A should only be used when staining the endogenous protein.

Fig2B; it is not clear when the samples for WB were taken, and the reader would like to know if the residual colonies that arise in the CENP-A- deleted setting that are not reconstituted with CENP-A or the MT1-CH3 mutant are due to residual expression of endogenous CENP-A.

Fig4. G,H; y-axis is labeled "% of multipolar cells" this is a little confusing in relation to Fig.4C "% of cells with multipolar spindles". I suggest changing this to "fraction of multipolar cells with a

given number of centrioles" or something similar.

Reviewer #3 (Remarks to the Author):

Centromere is specified by sequence independent epigenetic mechanisms and the CENP-A containing nucleosomes function as an important epigenetic marker for centromere specification. Although many kinetochore components assemble on the CENP-A containing chromatin, it is still unclear how kinetochore components recognize the CENP-A chromatin. In this paper, authors demonstrated that CENP-A N-terminus is -amino tri-methylated by NRMT. Furthermore, they showed that -amino tri-methylation of CENP-A is critical for recruitment of kinetochore protein CENP-T and CENP-I. Finally, they investigated defect of -amino tri-methylation of CENP-A in various cells and methylation-dead CENP-As show severe phenotype in p53-deficient HCT116 cells.

This MS contains new findings and would be interesting to general readers. But, later parts of this study using several cell lines are a bit unclear. It would be better if authors focus on some of important discoveries in this study. Substantial revision would improve quality of the paper. My specific concerns are following.

1. For IF experiments with anti-amino tri-methylation antibody, authors used cells expressing an eGFP-tagged CENP-A for main Figure. Although they showed some data for endogenous CENP-A with anti-amino tri-methylation antibody in Supplemental Figure, it would be better to use endogenous CENP-A. If detection efficiency is different between endogenous and exogenous CENP-A, they should describe this point.
2. Related with this point, How much population of CENP-A is amino tri-methylated? In addition, non centromeric CENP-A or soluble CENP-A with H4-HJURP are methylated? I feel that these are interesting points.
3. Concerning RPE1 replacement experiments, although MT1-CENP-A mutant rescued CENP-A deficiency (Figure 2), CENP-T and CENP-I levels were significantly decreased in both MT1-CENP-A cells and MT1-CENP-A-CH3 cells. I feel that reduction level of these proteins in MT1-CENP-A-CH3 cells are a little higher than those in MT1-CENP-A cells. Is this correct? In addition, why RPE-1 cells with MT1-CENP-A are viable even if CENP-T and CENP-I levels were significantly decreased? Please clarify these points.
4. For HeLa and HCT116 cells experiments, authors focused on spindle phenotype. Why they did not demonstrate level of centromeric proteins when MT-CENP-A mutants were replaced with wild-type CENP-A? It is a little odd to discuss about spindle phenotype without data for CCAN assembly. CENP-T and CENP-I were decreased in HeLa and HCT116 cells expressing MT-CENP-A mutants like RPE-1 cells? Did they observe some differences between various cell lines?
5. In course of HeLa and HCT116 cells experiments, they used MT-CENP-A mutants (not MT-CENP-A-CH3 mutants). Do they see more severe defects if they used MT-CENP-A-CH3 mutants?
6. They discussed Force imbalance in HCT 116 cells (p53 minus plus MT-CENP-A mutant). I am not sure that this caused by kinetochore defects.
7. I think that Figure 7 experiments are a bit out of scope of this paper. They should focus on kinetochore architecture in cells expressing MT-CENP-A mutants. Cancer phenotype (and spindle phenotype) should be reported in separate paper. They can focus on phenotype analysis of CCAN assembly and can publish this as a report format. (later parts of this study are unclear for me.)
8. N-terminal CENP-A is usually not conserved among different organisms. This modification may be human specific. It would be better to mention evolutionary view points for this modification in Discussion.

Minor points

1. Figure 3E, merge means CENP-I + CENP-B?
2. 0.1m nocodazole should be 0.1M nocodazole.

Reviewer I

1.) In both the introduction and the discussion, it is stated that RCC1 is the only NRMT1 substrate for which the impact of N-terminal methylation is known. This is untrue, and authors should include papers showing the impact of N-terminal methylation on DDB2 and CENP-B function (Cai et al, JBC, 2014; Dai et al., Journal of Proteosome Research, 2013)

We appreciate the reviewer's point regarding the importance of citing these papers. In the revised manuscript we specifically reference the work by Cai et al. and Dai et al. page 4 in elucidating the functional importance of amino-terminal methylation for these proteins.

2.) Authors state that the CENP-A mutants MT1 and MT2 are recognized by an antibody against phospho-Ser16/18 (Figure 1J). However, MT2 is not included in this figure, and the Ser16/18 phospho status of MT3 is not mentioned. These data should be included.

In the revised manuscript we present data showing the both MT2 and MT3 are phosphorylated similar to wild-type CENP-A, fully supporting the idea that CENP-A methylation and phosphorylation at S16;S18 are independent events. These new data are shown in supplementary figure S2C.

3.) Figures 3B and D are meant to demonstrate a "significant" reduction of CENP-C levels in cells rescued with WTCENP-ACH3. However, this does not appear to be the case by eye (Figure 3B) and there is no denotation of significance in the quantification (Figure 3D). If significance is to be claimed, the proper statistics will have to be shown and explained. Authors also claim a "significant" reduction in CENP-T and CENP-I (Figures 3C and F) without use of statistics. This also needs to be rectified.

P values indicating statistical significance are included for data presented in figure 3 of the revised manuscript. These tests show that loss of methylation of CENP-A coincides with a statistically significant reduction CENP-T and CENP-I at centromeres. Consistent with the previous report by Fachinetti et al., the CENP-A-H3 showed a significant reduction in the recruitment of CENP-C. However, the lack of methylation did not show as significant effect on CENP-C.

4.) It has long been hypothesized that N-terminal methylation could regulate protein-protein interactions, but there is little experimental proof. While Figure 3 hints at this, the impact of this paper could be significantly enhanced if the authors could directly demonstrate that unmethylated recombinant CENP-A (or mutant) binds less CENP-T or CENP-I.

We agree that knowing whether the recruitment of CENP-T and CENP-I are due to a direct interaction with the methylated CENP-A tail is potentially interesting. This is one

of several interesting possible explanations for the altered CENP-T/I recruitment. We attempted to conduct peptide pulldown experiments from interphase and mitotic extracts using methylated and unmethylated peptides (see figure below). Although we observed some increase in CENP-T and CENP-I in the methylated peptide, the results were highly variable and never robust enough for us to be confident that the assay is reliable; therefore, we did not include these data in the manuscript. That we do not see a clear preference for methylated CENP-A in the peptide pulldown may suggest that the direct interaction is not the primary mediator of CENP-I & T recruitment, and perhaps points to a less direct but potentially as interesting effect on the surrounding CENP-A nucleosome organization. However, solving this question is beyond the scope of work we are able to complete for these revisions.

5.) Figures 7D and E are labeled tumor free survival. As this term is usually meant to denote the length of time after primary treatment that an animal goes without signs of recurrence, it does not apply here. It is unclear if authors are measuring survival after injection or number of days until tumor appearance, but graphs should be relabeled to make this clear. The Materials and Methods section state tumor volumes were measured, this data should also be included, as the premise is that loss of methylation increases growth. The legend for Figure 7 does not match the figure.

We thank the reviewer for the comments and we have altered the graphs in figure 8 G&H (previously Figure 7D&E) to read “Tumor incidence”.

6.) Some typos in figure legends. Last bar of Figure 2D is labeled CENP-A-/-, where everywhere else it is CENPA-/F. Supplemental Figure 2 refers to Ser7, while the text refers to it as is Ser6. Supplemental Figures 3A and D are labeled + AdCre but are supposed to contain endogenous CENP-A.

Thanks to the reviewer for bringing these errors to our attention. The revised manuscript is edited accordingly. Please note that we retained the CENP-A -/- in figure 2D because the cells have been treated with Ad-CRE and are therefore now CENP-A null.

Reviewer II

1. The authors suggest that the defects in chromosome alignment and spindle integrity are due to reduced recruitment of the CCAN complex. Most notably, they see reduction in recruitment of CENP-T and CENP-I. To nail this issue, they should show that reduction in CENP-I and/or -T levels, independent of alterations in CENP-A methylation cause a similar phenotype, and moreover, show that this phenotype can be rescued if CENP-I or -T is recruited in a methylation-independent manner. Admittedly, the latter

experiment might not be possible if the reduction in CENP-I and -T occur independent from each other.

In response to the reviewer's suggestion, we suppressed both CENP-T and CENP-I in HeLa-TREx and HCT116p53^{-/-}, HCT116p53^{+/+} cells and examined the affect of this suppression on multipolar spindle formation. The suppression of CENP-T resulted in significant increase in multipolarity in p53 negative cells. This is similar to what was reported by Kevin Sullivan's lab for the reduction of the CENP-T binding partner CENP-W (Kaczmarczyk and Sullivan 2014). CENP-I did not cause a similar increase in multipolirity; therefore, we conclude that multipolarity defects in the CENP-A methylation mutants result from loss of the CENP-T/W complex.

2. The authors very nicely show that the multipolar spindle phenotype that is observed upon perturbation of CENP-A methylation is due to a force imbalance. They fail to provide an explanation for this imbalance, and also fail to provide an explanation for the very interesting observation that the force imbalance is not apparent (or not causing a phenotype) in p53-proficient cells. I assume that the force imbalance is due to a reduction in K-fiber strength, but the fact that segregation errors are equally frequent in p53-proficient and -deficient cells suggests that K-fibers are affected in both settings? Does weakening kinetochore function in the same p53-proficient and deficient cells result in a similar difference?

We agree that the cause of the force imbalance is not immediately obvious. In the revised manuscript we demonstrate that NDC80 recruitment is decreased when CENP-A cannot be methylated. This is consistent with role of CENP-T in NDC80 recruitment demonstrated by others. These data are presented in Figure 3 L&M and provide a possible mechanism by which CENP-A methylation defects induce defects in the MT spindle. Moreover, in response to the reviewer's comments, we show that CENP-A methylation mutants (^{MT1}CENP-A) can produce multipolar spindles in RPE non-transformed cells when p53 is suppressed. Showing that p53 activity is the essential difference between HCT116 p53 wt and ^{-/-} cells that accounts for the presences of multipolar spindles in these experiments.

The authors very nicely show that the growth of tumor cell lines is enhanced in tissue culture and in mice, when CENP-A methylation and p53-function are perturbed. But it remains speculative if this is directly linked to the mitotic defects described in the first part of the paper. The authors should discriminate if proliferation rates are affected, or if the enhanced size of colonies and tumor is due to enhanced survival of the cells under these conditions. Given that they report an increase in segregation errors, one would expect that effects on tumor growth are not a direct effect on proliferation, but rather caused by the evolution of more aggressive cell clones (than could of course proliferate faster). Do the colonies/tumors display heterogeneity in chromosome numbers, as could be expected?

In response to the reviewer's comments we looked more closely at proliferation rates of the MT1-CENP-A and WT-CENP-A expressing HCT116 cells and noted that the population of MT1-CENP-A expressing cells did have an increased rate of proliferation. However, both the colony formation assays (Figure 8B&C) and the newly conducted Alamar Blue assays (Figure 8E&F) indicate that the increased proliferative potential is not uniform across the entire MT-CENP-A expression population, but that a subset of cells acquire a more proliferative phenotype, whereas most cells grow similarly to WT-CENP-A expressing cells. However, we did not see enhanced survival of the cells after sorting single cells into 96 well plate (Fig. 8E). We looked to see whether the more proliferative cells had abnormal chromosome number, however, we did not observe a difference between these cells and WT-CENP-A expressing cells.

Does partial inhibition of Eg5 affect colony size in the p53-deficient cells (and not in the p53-proficient cells)?

We appreciate that the reviewer is interested to know whether the multipolar spindle defect is contributing the differences in colony size observed in the p53^{-/-} HCT116 cells. However, since Eg5 is involved in interphase processes of the cell (including protein synthesis), even if cells survived long-term treatment with Monastrol, we were not confident that we would be able to attribute effects on colony size to spindle bipolarity rather than an other functions of Eg5.

Minor points:

Fig.1; blots are consistently labeled with me3-CENP-A, but shouldn't this be me3-CENP-A-GFP? Me3-CENP-A should only be used when staining the endogenous protein.

In accordance with the reviewer's suggestion, the blots in figure 1 have been relabeled as me3-CENP-A-eGFP to denote the staining of the expressed GFP-tagged CENP-A rather than the endogenous CENP-A protein.

Fig2B; it is not clear when the samples for WB were taken, and the reader would like to know if the residual colonies that arise in the CENP-A- deleted setting that are not reconstituted with CENP-A or the MT1-CH3 mutant are due to residual expression of endogenous CENP-A.

The western blots shown in Figure 2B are blots of cells prior to Ad-cre infection to demonstrate the level of CENP-A overexpression in these cells. When the extracts were made relative to Ad-Cre infection is noted in the figure legend of the revised manuscript. The number and percentage of residual colonies observed both in CENP-A^{-/-} parental cells and the ^{MT1}CENP-A^{H3} expressing cells are nearly identical. Since we know that CENP-A is essential, and that viable colonies seen in previous experiments by Fachinetti et al. were due to incomplete removal of the Floxed CENP-A allele, we think that the most likely explanation is that the viable colonies in ^{MT1}CENP-A^{H3} condition are

failed to remove the CENP-A floxed allele. Unfortunately it is not possible to look at the endogenous CENP-A status of the colonies in Figure 2 retrospectively. Moreover, Ad-Cre treatment itself can cause cell viability defects, and so we were careful to titrate the Ad-Cre concentration to avoid these effects; however this meant that a low level of recombination failure (approximately 12 %) was observed.

Fig4. G,H; y-axis is labeled "% of multipolar cells" this is a little confusing in relation to Fig.4C "% of cells with multipolar spindles". I suggest changing this to "fraction of multipolar cells with a given number of centrioles" or something similar.

The axes of Figure 4 G & H were re-labeled as "Percent cells per centriole number".

Reviewer III

1. For IF experiments with anti-amino tri-methylation antibody, authors used cells expressing an eGFP-tagged CENP-A for main Figure. Although they showed some data for endogenous CENP-A with anti-amino tri-methylation antibody in Supplemental Figure, it would be better to use endogenous CENP-A. If detection efficiency is different between endogenous and exogenous CENP-A, they should describe this point.

We appreciate the reviewer's point! Methylation of endogenous CENP-A is clearly lost in the NRMT1 shRNA treated cells, similar to what we observed in cells expressing CENP-A-GFP. In the revised manuscript we include anti-methylated CENP-A staining in both parental and CENP-GFP expressing cells in the main figure set (Figure 1) in response to the reviewer's request.

2. Related with this point, How much population of CENP-A is amino tri-methylated? In addition, non centromeric CENP-A or soluble CENP-A with H4-HJURP are methylated? I feel that these are interesting points.

We agree the degree of CENP-A methylation in the pre-nucleosomal and nucleosomal populations are an interesting aspect of this modification. However, we feel that we have adequately addressed these issues using quantitative Mass spectrometry approaches in a previous paper published in PNAS (Bailey et al. 2013). We found that about one third of CENP-A is methylated when it is bound to HJURP during mitosis, in the pre-nucleosomal form. Seventy-five percent of nucleosomal CENP-A from asynchronously growing HeLa cell cultures was methylated, and 90% of nucleosomal CENP-A in mitosis was methylated.

3. Concerning RPE1 replacement experiments, although MT1-CENP-A mutant rescued CENP-A deficiency (Figure 2), CENP-T and CENP-I levels were significantly decreased in both MT1-CENP-A cells and MT1-CENP-A-CH3 cells. I feel that reduction level of these proteins in MT1-CENP-A-CH3 cells are a little higher than those in MT1-CENP-A cells. Is this correct? In addition, why RPE-1 cells with MT1-CENP-A are viable

even if CENP-T and CENP-I levels were significantly decreased? Please clarify these points.

Quantitation from over 500 centromeres in figure 3C and 3F clearly shows a reduction of CENP-T and CENP-I that is dependent on methylation of the amino terminus, irrespective of the c-terminus. However, the reviewer is correct that swapping the CENP-A C-terminus for that of histone H3 also reduces CENP-T and CENP-I recruitment. This is likely due to the loss of CENP-C. Regardless for the case of CENP-I and CENP-T the largest reduction is seen when the C-terminus is intact suggesting that the methylation is the primary contributor to CENP-I and CENP-T accumulation. However, we think that the levels of CENP-T and CENP-I recruited through the c-terminus (via CENP-C) are sufficient to sustain cell viability. Therefore we only see the defect in cases where CENP-T and CENP-I are most severely affected, i.e. ^{MT1}CENP-A^{H3} expression.

4. For HeLa and HCT116 cells experiments, authors focused on spindle phenotype. Why they did not demonstrate level of centromeric proteins when MT-CENP-A mutants were replaced with wild-type CENP-A? It is a little odd to discuss about spindle phenotype without data for CCAN assembly. CENP-T and CENP-I were decreased in HeLa and HCT116 cells expressing MT-CENP-A mutants like RPE-1 cells? Did they observe some differences between various cell lines?

The reviewer is correct that in the original manuscript the multipolar spindle phenotype, and CENP-T and CENP-I reductions were shown in different cell types. In the revised manuscript we analyzed the role of CENP-A methylation on the recruitment of CENP-I and CENP-T to centromere in HeLa-Trex (Fig.S3I&J) cells in addition to the RPE cells that were part of the original manuscript. We observe that, similar to RPE cells, HeLa-Trex cells expressing the ^{MT1}CENP-A construct show a reduced recruitment of CENP-I and CENP-T compared to the wild-type CENP-A, when endogenous CENP-A was suppressed by shRNA (Fig.S3I&J). In addition, we show that the multipolar spindle defects occur in RPE cells expressing ^{MT1}CENP-A when p53 is suppressed. Therefore the revised manuscript shows a complete concordance in two cell types between the reduction of CENP-I and CENP-T and formation of multipolar spindles dependent on the loss of CENP-A methylation.

5. In course of HeLa and HCT116 cells experiments, they used MT-CENP-A mutants (not MT-CENP-A-CH3 mutants). Do they see more severe defects if they used MT-CENP-A-CH3 mutants?

We expect that the worsened phenotype observed with MT1-CENP-A-CH3 compared with MT1-CENP-A is attributable to the loss of CENP-C recruitment. However, for the purposes of this manuscript we focused primarily on defects that were directly attributable to the methylation status of CENP-A we restricted our analysis in HCT116 to the MT1-CENP-A constructs. The revised manuscript demonstrates that the effects of p53 reduction on CENP-A methylation mutant-induced multipolarity occur in RPE cells,

as well as HCT116. And the reductions in CENP-T and CENP-I upon MT1-CENP-A expression we observed initially in RPE cells held true in HeLa cells. Together these new observations suggest that the alterations we are observing in response to MT1-CENP-A are common between different cell types.

6. They discussed Force imbalance in HCT 116 cells (p53 minus plus MT-CENP-A mutant). I am not sure that this caused by kinetochore defects.

In the revised manuscript we show that NDC80 levels are reduced when CENP-A is unable to be methylated. NDC80 disruption can lead to multipolar spindles through a similar mechanism of centriole disengagement (Leber et al. 2010) in some cell types. Likewise, expression of a dominant negative GFP-NDC80 leads to pole splitting (Mattiuzzo et al. 2011). Therefore, NDC80 loss, in part, may explain how CENP-A methylation leads to multipolarity.

7. I think that Figure 7 experiments are a bit out of scope of this paper. They should focus on kinetochore architecture in cells expressing MT-CENP-A mutants. Cancer phenotype (and spindle phenotype) should be reported in separate paper. They can focus on phenotype analysis of CCAN assembly and can publish this as a report format. (later parts of this study are unclear for me.)

At the suggestion of the editor we have retained figure 7 in the revised manuscript. While we appreciate that the much of the manuscript is focused on mechanistic understanding of CENP-A methylation, we feel that understanding CENP-A misregulation in the context of the tumorigenesis is essential to the field.

8. N-terminal CENP-A is usually not conserved among different organisms. This modification may be human specific. It would be better to mention evolutionary view points for this modification in Discussion.

The reviewer is correct that the amino acid sequence of the amino terminus of CENP-A is highly divergent across species. And while we have not directly addressed the conservation of CENP-A methylation, previous work from Ian Macara's and Don Hunt's groups showed that peptides containing the sequences identical to the amino termini of cows, mice, zebrafish and chickens are readily methylated by NRMT (Petkowski et al. 2012). This strongly suggests that CENP-A methylation is conserved in these species. We mention this work in the discussion of the revised manuscript (page 19).

Minor points

1. Figure 3E, merge means CENP-I + CENP-B?

The specific merged channels if figure 3B and E are noted on the revised figure.

2. 0.1m nocodazole should be 0.1M nocodazole.

The value for nocodazole was corrected to 0.1 micromolar.

REVIEWERS' COMMENTS:

Reviewer #1 (Remarks to the Author):

The authors have satisfactorily addressed my comments, I recommend the manuscript for publication.

Reviewer #2 (Remarks to the Author):

The authors haven't exactly addressed all of the points that I raised, but nonetheless I would recommend publication. Their demonstration that depletion of CENP-T reproduces the phenotype seen when CENP-A cannot be methylated is a very nice addition, as well as the demonstration that depletion of p53 in RPE's under conditions where CENP-A cannot be methylated also causes the spindle to become multipolar. I agree with the authors that continued culture in monastery, is unlikely to produce viable cells, but what I intended was for the authors to restore spindle bipolarity by a short-term exposure to Eg5 inhibitors, this would provide some more insight in the force imbalance. Such experiments have been executed with success in the past (see van Heesbeen et al., Cell Rep. 2014). But this can be addressed in future experiments

Reviewer #3 (Remarks to the Author):

Authors addressed concerns from reviewers and the MS has been much improved. I do not have additional comments on the revised MS.